# ARE OBJECT-CENTRIC REPRESENTATIONS BETTER AT COMPOSITIONAL GENERALIZATION?

## ABSTRACT

Compositional generalization – the ability to reason about novel combinations of familiar concepts – is fundamental to human cognition and a critical challenge for machine learning. Object-centric learning, representing a scene as a set of objects, has been proposed as a promising approach for achieving this capability. However, systematic evaluation of these methods in visually complex settings remains limited. In this work, we introduce a Visual Question Answering benchmark consisting of three different visual worlds to measure how well vision encoders, with and without object-centric biases, generalize to unseen combinations of object properties. To ensure a fair and comprehensive comparison, we carefully account for the capacity of the image representation, training data diversity, downstream compute, and sample size. In this study, we use DINOv2 and SigLIP2, two widely used vision encoders, as the foundation models and their object-centric counterparts. Our key findings reveal that (1) object-centric approaches are superior in harder compositional generalization settings; (2) original dense representations surpass OC only on easier settings and typically require substantially more downstream compute; and (3) OC models are more sample-efficient, achieving stronger generalization with fewer images, whereas dense encoders catch up or surpass them only with sufficient data and diversity. Overall, object-centric representations offer stronger compositional generalization when any one of training data diversity, sample size, or downstream compute is constrained.

## 1 INTRODUCTION

Compositionality, the ability to perceive and generalize to novel combinations of familiar elements, is widely seen as a cornerstone of human cognition and has long been linked to the systematic ability of humans to understand and produce novel expressions from known parts (Fodor & Pylyshyn, 1988; Treisman, 1996; Chomsky, 2002). In machine learning, *compositional generalization* – the robustness of models to novel combinations of familiar concepts – has been explored in various forms. In natural language, compositional generalization can be assessed by testing the response of the model to rearranged or recombined words and numbers (Lake & Baroni, 2018; Dziri et al., 2024); in vision, it may involve creating novel objects by recombining seen object properties or combining known objects in novel configurations (Kim et al., 2024; Haramati et al., 2024; Montero et al., 2024; Abbasi et al., 2024).

Even though modern VLMs and generative models show impressive abilities, multiple studies have shown that they remain brittle on rigorous tests for compositional generalization. For instance, ConMe exposes a *benchmarking gap*, inducing up to a 33% drop in accuracy for state-of-the-art VLMs once negatives are made genuinely hard (Huang et al., 2024). SugarCrepe demonstrates that several widely used "compositionality benchmarks" were hackable – so much so that *blind* (image-free) models could outperform vision–language systems, revealing spurious lexical cues and weak binding of attributes and relations (Hsieh et al., 2024). In text-to-image, ConceptMix and GenAI-Bench both find steep degradations as prompts combine more entities, attributes, spatial relations, or logic; models frequently omit objects, misbind attributes, or miscount (Wu et al., 2024; Li et al., 2024). Scaling alone has not solved this: recent studies report large accuracy drops on unseen combinations despite substantial data increases, and show that compositional generalization strongly depends on pretraining frequencies and diversity (Kempf et al., 2025; Wiedemer et al., 2025; Uselis et al., 2025).

These observations have motivated vision representations aimed at supporting compositional generalization more naturally. In particular, *object-centric (OC) representations* represent a scene as a collection of objects, commonly binding different objects into separate *slot vectors* (Locatello et al., 2020). Because such representations match the natural structure of a scene by decomposing it into discrete objects, they are conjectured to provide more compositional and generalizable representations (Greff et al., 2020; Locatello et al., 2020; Dittadi et al., 2022; Jiang et al., 2023; Brady et al., 2023). However, beyond a few preliminary indications (Yoon et al., 2023; Montero et al., 2024; Kim et al., 2024; Haramati et al., 2024), the relationship between object-centric representations and compositionality remains largely untested in a systematic and principled manner. In this work, we investigate those claims in greater depth. Specifically, we study how well different visual representations support compositional generalization of object properties on the visual question answering (VQA) task.

The flavor of compositionality we are most interested in is *object property composition* (Johnson et al., 2017; Abbasi et al., 2024; Montero et al., 2024; Kim et al., 2024) – the ability of a model to generalize to novel combinations of previously seen object properties. For example, a model trained only on red cubes and blue spheres should be able to successfully handle blue cubes at test time. As this form of compositionality requires precise control over the factors of variation in the visual world, most works rely on synthetically generated images from a computer graphics tool (Kim et al., 2024; Montero et al., 2024) or a pretrained generative model (Abbasi et al., 2024). Although compositionality is often described as a core motivation for object-centric representations, its evaluation is typically limited to changing the number of objects (Johnson et al., 2017; Locatello et al., 2020; Karazija et al., 2021; Biza et al., 2023). The works most similar to ours are Kim et al. (2024) and Montero et al. (2024), both investigating compositional generalization of object properties. However, Kim et al. (2024) use a protocol that only allows evaluation of generative models rather than general image representations and do not isolate which design choices contribute to better performance, while Montero et al. (2024) examine compositionality only under the more limited setting of simpler images with a single object.

In order to rigorously study the compositional generalization capabilities of visual representations for object property composition, we design our own benchmark. First, inspired by Kim et al. (2024), we generate images in a CLEVRTex-(Karazija et al., 2021), Super-CLEVR- (Li et al., 2023), and MOVi-C-style (Greff et al., 2022), allowing us to precisely define the entire visual world. Specifically, we consider every combination of individual factors of variation (e.g., shape, material, and size) characterizing each object. Then, we reserve 20% of these object combinations for testing compositional generalization while allocating the rest to progressively smaller subsets for training. This ensures that no test objects of the compositional generalization dataset were encountered during training, even though their individual properties were. To evaluate this compositional generalization via VQA, we follow Mamaghan et al. (2024); Li et al. (2023) by generating question–answer pairs for all images. This results in three different base datasets with 3 training datasets each – CLEVRTex, Super-CLEVR, and MOVi-C *"easy"*, *"medium"*, and *"hard"* – and one dataset for each base dataset dedicated to testing compositional generalization, called *"COOD"*.

For our comparisons, we focus on pretrained foundation models and object-centric models that incorporate such foundation models as backbones, a leading approach in this domain. Specifically, we use DINOv2 (Oquab et al., 2023) and SigLIP2 (Tschannen et al., 2025) as the foundation models with DINOSAURv2 (Seitzer et al., 2022; Didolkar et al., 2024) and SigLIPSAUR2 as its object-centric counterparts. To ensure a fair and comprehensive comparison, we account for differences in representation format by controlling for image representation sizes, both for the number of tokens and the token dimension, ensuring that differences in compute allocation do not unfairly advantage one approach over another. We evaluate all models by training distinct downstream models on the VQA task on training sets of increasing difficulty, testing on in-distribution (ID) as well as compositional out-of-distribution (COOD) generalization sets. Following the framework of Mamaghan et al. (2024), we vary the size of the downstream model and, additionally, the input size of the image representation. Finally, by carefully controlling the visual combinations that models are exposed to at train and test time, we can systematically adjust the hardness of the generalization task until even an oracle with access to ground-truth inputs struggles to generalize at test time.

Our main contributions can be summarized as follows:

- **Datasets**: We design our own compositional generalization benchmark based on the CLEVRTex, Super-CLEVR, and MOVi-C (images) datasets (Karazija et al., 2021; Li et al., 2023; Greff et al., 2022; Kim et al., 2024; Mamaghan et al., 2024). To assess compositional generalization, we

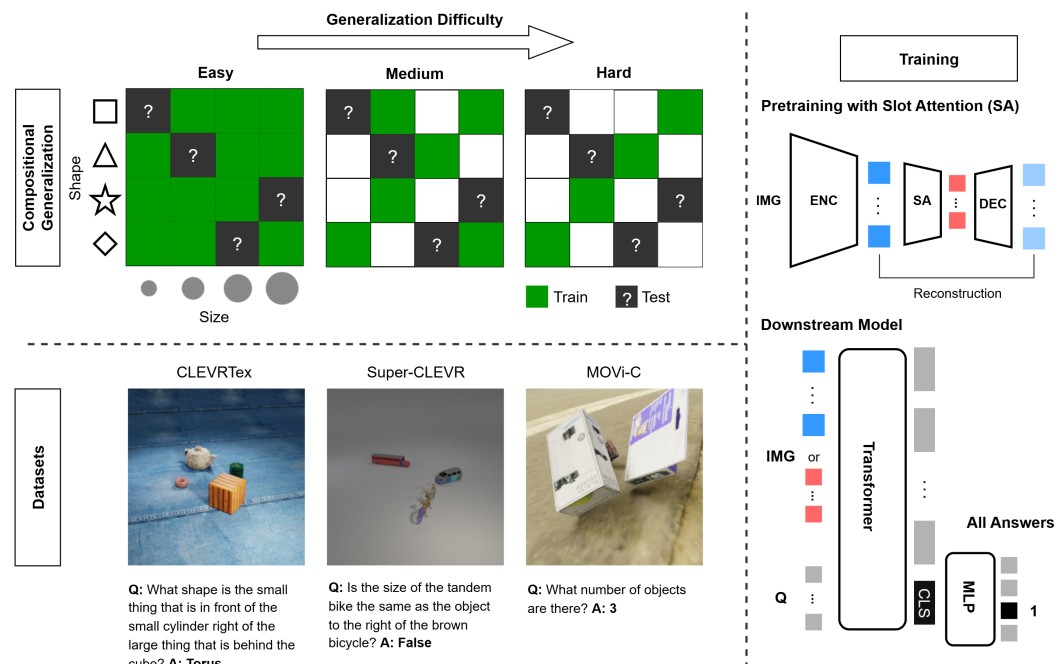

Figure 1: **Compositional Generalization.** To increase generalization difficulty, we decrease the number of unique object property combinations that are seen during training. In the conceptual example, each object is defined by its shape and size, which coincides with MOVi-C. **Datasets.** For each generalization difficulty and base dataset, we generate images and corresponding question–answer pairs by sampling objects with the allowed combinations. **Training.** We pretrain object-centric (OC) models by reconstructing the self-supervised (Dense) features from pretrained vision encoders. For VQA downstream training, we concatenate the image features (OC: red; Dense: blue) with the fixed text embeddings and train transformer models of various sizes to predict the answer given image and question.

define fixed held-out test sets containing 20% of all object-property combinations, along with three progressively smaller subsets from the remaining combinations. The smaller the subset, the greater the challenge for generalization. The compositional test set ensures that no test objects appear during training, while all their individual properties – such as e.g., shape, material, and size – are encountered. Finally, we generate question–answer pairs for all images to evaluate all models on a VQA task.

- **Finding I (Training diversity).** Reducing training diversity inflates ID accuracy but hurts COOD; object-centric (OC) representations degrade *less* and remain superior to dense features on harder generalizations at comparable representation sizes. With maximal diversity, OC stays competitive; dense features only match or slightly surpass OC on the *easiest* settings, and typically require stronger downstream models to do so (see §4.1,§4.2).

- **Finding II (Compute).** At matched downstream FLOPs, OC representations deliver higher COOD across budgets; they are especially strong with smaller downstream models. Dense counterparts need substantially more compute (and often larger downstream models) to overtake OC – and then mainly on easier generalizations. Cross-attention resizing often offers no compute–accuracy benefit: downsizing dense features via CA remains worse than equally small OC features, and upsizing OC via CA consistently hurts the performance (§4.3).

- **Finding III (Sample size).** OC is more sample-efficient: it reaches better COOD with fewer images, particularly with a small downstream model. Given *enough* data *and* a larger downstream model, dense features can match or slightly surpass OC on the easiest settings; otherwise OC remains better. Increased data diversity amplifies the COOD gains from larger sample sizes, where the benefit of adding more samples of less diverse data slows down or even goes down (§4.4).

## 2 RELATED WORK: COMPOSITIONALITY

This section briefly summarizes different ways in which compositionality has been defined and tested.

**Text.** Lake & Baroni (2018) studied compositionality by training a model to decode natural-language commands into action sequences that feature novel combinations of concepts at test time. Dziri et al. (2024) demonstrated that transformers can fail catastrophically on seemingly simple tasks (e.g., multi-digit integer multiplication) when test conditions differ slightly from training (e.g., more digits).

**Images.** Kim et al. (2024) explored compositionality without language annotations by constructing a visual world of objects with simple attributes (e.g., shape, texture). They controlled which portion of the combinatorial attribute space was shown during training and formulated a generative task where the model must learn and apply transformation rules (e.g., swapping shapes) to unseen combinations at test time. Haramati et al. (2024) probe, among other things, the compositional generalization of different components of their architecture in a reinforcement learning task that involves arranging objects in a specified way on a table with a robotic arm.

**Text-to-image.** Some recent work frames compositionality as a text-to-image generation task, prompting models with increasingly complex combinations of visual concepts to test that all mentioned concepts appear in the generated image (Wu et al., 2024; Li et al., 2024).

**Image-to-text and VQA.** The *SugarCrepe* benchmark evaluates compositional comprehension by presenting an image alongside a correct caption and a closely matched "hard negative", which can involve object swapping or replacement (Hsieh et al., 2024). The model must choose the caption that accurately describes the image, extending earlier approaches such as Ma et al. (2023).

**Object-centric representations.** In the context of reinforcement learning, Yoon et al. (2023) and Haramati et al. (2024) found that object-centric representations are mostly beneficial for tasks requiring relational reasoning with object interactions. Additionally, Haramati et al. (2024) also demonstrated that their agent can generalize compositionally to more objects than seen during training, both empirically and theoretically. Kim et al. (2024) provided some evidence that a slot-based State-Space Model improves compositional generalization, though the specific design elements driving this improvement remain unclear. Furthermore, Montero et al. (2024) show that a simple object-centric model reconstructs novel objects with hold-out ranges of properties (e.g., color or rotation) for a single object better than a non-object-centric alternative when the models have been trained on all combinations for the rest of the objects. Rubinstein et al. (2025) and Baldassarre et al. (2024) both advocate for revisiting the original goals of object-centric learning and a departure from the evaluation of these representations solely or mostly on (unsupervised) image segmentation. Concretely, their downstream tasks consist of OOD image classification or scene classification and action recognition in videos, respectively.

## 3 PROBLEM SETUP

### 3.1 DATASET GENERATION

Inspired by Kim et al. (2024), we create datasets in the style of CLEVRTex (Karazija et al., 2021), Super-CLEVR (Li et al., 2023), and MOVi-C (Greff et al., 2022). For each base dataset, we create training splits with progressively smaller subsets of all possible objects, resulting in increasingly harder OOD problems. For example, for CLEVRTex, each object is defined by a triplet of properties, shape, size, and material, yielding 192 unique objects in total (see Appendix A for details about the other base datasets). We then randomly select 3–6 objects from the set of allowed objects per scene. We render images using Blender[1]. As a result, we obtain three training datasets per base dataset – CLEVRTex, Super-CLEVR, and MOVi-C *"easy"*, *"medium"*, and *"hard"* – each time decreasing the diversity by roughly halving the number of admissible objects. Every training set consists of 48k images: 40k for training, 4k for validation, and 4k for in-distribution testing. Finally, we generate a COOD test set for each base dataset, each containing 4k images using the remaining 20% of objects.

Our goal is to evaluate the quality of representations using VQA. Thus, for each image, we generate multiple question-answer pairs, using the generation approach of Johnson et al. (2017) adapted to

---

[1] https://www.blender.org/

CLEVRTex and MOVi-C, and the existing implementation for Super-CLEVR (Li et al., 2023) (see Appendix A). This results in 42 question-answer pairs per image on average, resulting in roughly 1.7M per training set and 170k per test set (ID & COOD).

## 3.2 Models and Evaluation

**Setup.** To evaluate VQA, we follow the setup of Mamaghan et al. (2024). The downstream VQA model is a transformer that receives concatenated text and image representations as input and outputs a class label (details in Appendix B.3). We report results for two different sizes: a small 2-layer variant (TF 2) and a larger 5-layer variant (TF 5). Questions are encoded by a pretrained T5-base model (Raffel et al., 2020). The answers are represented as 28 (CLEVRTex), 106 (Super-CLEVR), or 48 (MOVi-C) distinct labels, which include "yes", "no", natural numbers up to the maximum number of objects, and all possible values of object properties (including part names for Super-CLEVR).

To gauge dataset difficulty, we train two additional baselines: a naive question-only baseline using only the questions as inputs to the downstream model and a ground-truth oracle that supplies the true object properties of all visible objects in the scene as "image representations" for the downstream model. After training, each downstream model is evaluated on its corresponding in-distribution (ID) and compositional out-of-distribution (COOD) test set at every training checkpoint.

**Vision Models.** First, we evaluate the dense representations of two strong pretrained vision models: DINOv2 ViT-S/14 (Oquab et al., 2023) and SigLIP2 ViT-B/16 (Tschannen et al., 2025). We then consider different approaches of transforming these representations to study how COOD performance is affected. In particular, we pretrain an *object-centric model* for every dataset variant by reconstructing the pretrained dense representation with a Slot Attention (Locatello et al., 2020) bottleneck (Seitzer et al., 2022). This yields DINOSAURv2 (Didolkar et al., 2024) and, to the best of our knowledge, the first object-centric SigLIP2 variant, *SigLIPSAUR2*. Architectural and hyperparameter details are in Appendix B.1. As an alternative to Slot Attention, we also run k-means on the set of dense patch tokens to extract a set of cluster centroids representing the image (as Baldassarre et al. (2024)).

The original vision encoders and their respective object-centric counterparts produce image representations of different sizes, which strongly impacts the downstream model's FLOPs (see Appendix C for details). Concretely, in our setting, the number of tokens and token dimensions are: $[256, 384]$ for DINOv2 vs. $[7, 256]$ for DINOSAURv2, and $[196, 768]$ for SigLIP2 vs. $[7, 256]$ for SigLIPSAUR2. To enable a fair comparison, we change the size of the image representation with downstream model variants that include a single cross-attention layer immediately after the vision encoder output. This layer modifies the size of the image representation by using a number of learned queries matching the target size. We evaluate both increasing the size of the object-centric representation to the size of the original vision encoder, or, vice-versa, decreasing the representation size of the original vision encoder. The latter could be seen as a possible alternative to Slot Attention. The cross-attention layer is trained jointly with the downstream model, and results in identical compute requirements for these variants[2]. We also experimented with replacing the Slot Attention bottleneck, e.g., in DINOSAURv2, with a cross-attention module, and replacing the downstream model's cross-attention layer with Slot Attention. Both attempts yielded suboptimal results, suggesting either that more extensive tuning is needed or that these substitutions are ill-suited for the VQA task[3].

## 4 Experiments

**Summary.** Across three base datasets, CLEVRTex, Super-CLEVR, and MOVi-C, we study how training diversity, downstream compute, and sample size affect VQA compositional out-of-distribution (COOD) performance. First, in §4.1, we observe that decreasing training diversity increases generalization difficulty (Fig. 2). Then, comparing the generalization capabilities of different image representation types in §4.2, we observe that object-centric representations almost always match or surpass the original dense encoders when the generalization is sufficiently difficult (Tables 1 and 11). In §4.3, we consider downstream compute, which depends on the image representation size, and show

---

[2]We choose to ignore the compute from the cross-attention layer as the goal is to compare the performance of different representations fairly with respect to image representation size.

[3]For a discussion of cross-attention as an alternative to Slot Attention–and why it may be suboptimal for this kind of pretraining–see Wu et al. (2023).

Table 1: VQA accuracy (%) of both downstream models (TF 2 & 5) on the respective compositional generalization test sets for all DINOv2-based models trained on *"easy"* (E), *"medium"* (M), and *"hard"* (H) training sets. We compute deltas compared to the original pretrained vision encoder, and list the number of tokens of the representations ("size").

| | Model | Size | CLEVRTex | | | Super-CLEVR | | | MOVi-C | | |
| | | | E | M | H | E | M | H | E | M | H |
|---|---|---|---|---|---|---|---|---|---|---|---|
| **TF 2** | DINOv2 | 256 | 69.5 | 58.8 | 50.0 | 60.9 | 57.0 | 49.7 | 57.5 | 53.6 | 51.4 |
| | DINOv2 + CA | 7 | -1.2 | -4.8 | -2.3 | -1.2 | -1.2 | -0.8 | -1.7 | -0.1 | +0.2 |
| | DINOv2 + k-means | 7 | -16.5 | -9.1 | -3.1 | -10.1 | -7.1 | -2.2 | -5.8 | -2.6 | -1.6 |
| | DINOv2 + k-means | 128 | -1.1 | +1.2 | -0.6 | -1.4 | -0.8 | -0.1 | -0.7 | -0.3 | +0.9 |
| | DINOSAURv2 | 7 | +7.0 | +12.3 | +5.6 | -0.3 | +1.6 | +1.2 | +1.0 | +1.1 | +1.6 |
| | DINOSAURv2 + CA | 256 | +0.1 | +9.8 | +1.0 | -3.3 | -1.9 | -0.4 | -3.3 | -1.0 | -0.6 |
| **TF 5** | DINOv2 | 256 | 85.4 | 70.3 | 55.4 | 68.1 | 63.0 | 51.7 | 60.0 | 56.0 | 54.0 |
| | DINOv2 + CA | 7 | -6.5 | -2.4 | -1.7 | -2.9 | -1.9 | -0.9 | -1.7 | -0.7 | -0.8 |
| | DINOv2 + k-means | 7 | -32.6 | -21.3 | -9.1 | -17.5 | -13.1 | -4.5 | -8.5 | -6.0 | -4.9 |
| | DINOv2 + k-means | 128 | -4.7 | -4.3 | -1.3 | -3.7 | -1.2 | -0.4 | +0.6 | +0.5 | -0.1 |
| | DINOSAURv2 | 7 | -2.9 | +3.0 | +0.1 | -3.5 | -2.2 | +0.4 | -0.8 | -0.4 | 0.0 |
| | DINOSAURv2 + CA | 256 | -5.9 | -0.3 | -1.4 | -5.3 | -3.7 | -0.6 | -2.3 | -2.1 | -2.0 |

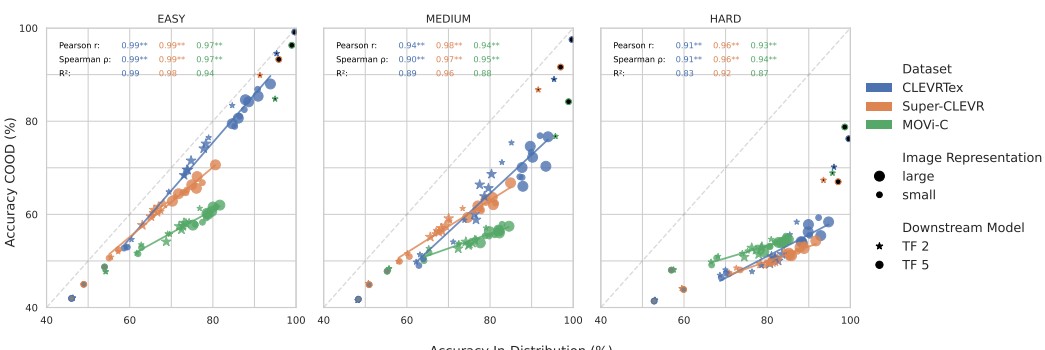

Figure 2: VQA in-distribution and compositional out-of-distribution accuracy are very strongly correlated (highly significant: p-value $< .01$). Performances for CLEVRTex, Super-CLEVR, MOVi-C *"easy"*, *"medium"*, and *"hard"* at the end of training with correlations and ground-truth oracle (upper right: filled black) and question-only baseline (lower left: filled gray).

that object-centric models are in general preferable both under constrained compute budgets and when the generalization is hard (Fig. 3). Finally, in §4.4, we elucidate the interplay of sample size and training diversity, and we find that OC models are more sample-efficient: dense representations only outperform them with enough diversity, sample size, and a larger downstream model (Figs. 4 and 5).

For easier readability, we often refer to a single base dataset in the following and explicitly mention if trends differ across datasets. All results can be found in Appendix D.

## 4.1 THE EFFECT OF DATA DIVERSITY

**Oracle.** We first validate that our experimental setup, including datasets and downstream models, is suitable for testing compositional generalization by establishing that a model with the "right" representation is able to solve the task in-distribution (ID) but still might lack in compositional out-of-distribution (COOD) generalization as the difficulty increases. Specifically, we train an oracle that uses the ground-truth object properties as image representation. For all training datasets, the oracle can achieve nearly perfect ID test accuracy (Fig. 2), given a sufficiently large downstream model. However, its compositional generalization drops notably when trained on smaller subsets of the full visual space. As an example, it still reaches almost 100% on CLEVRTex *"COOD"* by training on *"easy"*, but struggles to even reach 80% when training on *"hard"*.

**ID vs. COOD.** Having established the suitability of our setup, we now investigate models with learned image representations. Evaluating all vision encoders, as depicted in Fig. 2, we observe a consistent and intuitive pattern: as we constrain the diversity of the training data – thereby increasing generalization difficulty – the models' in-distribution accuracy improves due to fewer visual combinations to learn. However, this simplification in ID tasks simultaneously intensifies COOD challenges, as models must generalize from fewer learned combinations to the fixed COOD test set. This is consistent across all base datasets and vision encoders.

## 4.2 THE EFFECT OF IMAGE REPRESENTATION TYPE

**OC vs. Dense.** We naively compare the object-centric representations to their dense counterparts, ignoring the differences in their representation sizes. The object-centric versions are almost always better at compositional generalization with the smaller downstream model (Table 1). Concretely, the improvements in generalization for DINOSAURv2 over DINOv2 for TF 2 range from -0.3% to +12.3% (absolute) across all training datasets and are especially large on CLEVRTex. The trend is consistent across vision encoder families for SigLIP2-based representations (Table 11). When increasing the power of the downstream model (Table 1: TF 5), the dense representations are better for easier generalizations (*"easy"*) but lose their benefit when the OC representations either match or slightly surpass them for harder generalizations (*"hard"*). For example, the differences in generalization with the larger downstream model for DINOSAURv2 and its dense counterpart are from 0.0% to +0.4%. This is again consistent for SigLIP2-based models (Table 11).

**K-means vs. Slot Attention.** Comparing OC-like representations in Table 1, i.e., the pretrained Slot Attention and k-means variants, we observe that the k-means representations with the same number of tokens as the SA versions (7 tokens) are quite worse in compositional generalization. We hypothesize that this is due to the ineffectiveness of k-means, a method that does not use additional training, in drastically reducing the number of visual tokens (e.g., from 256 to 7 for DINOv2) by simply taking the centers of each cluster, compared to a "soft k-means" as performed by Slot Attention (Locatello et al., 2020). This is in contrast to Baldassarre et al. (2024), where a small number of tokens was often sufficient. This discrepancy may be explained by VQA being a task that requires more fine-grained visual information compared to the more global or coarse-grained tasks in Baldassarre et al. (2024). In order to partially overcome this, we increase the number of clusters used for k-means, i.e., the number of visual tokens here, to 128 such that there is still a reduction from the original representation while getting the best performance compared to using any number of fewer tokens (for details see Appendix B.2). The improved k-means representation, at the cost of using more tokens, is sometimes able to match the generalization capabilities of the SA versions with the bigger downstream model, especially on MOVi-C, while still falling behind for the smaller one (Table 1).

**Reduction with Cross-Attention.** To match the capacity of OC models, we turn to alternatives for decreasing the size of the original dense image representation. Using a cross-attention layer for this purpose, it is again not as effective as the SA models. Especially at the lower resource settings with a smaller downstream model, the downsized versions of DINOv2, i.e., DINOv2 + CA, are considerably worse compared to their object-centric counterparts in DINOSAURv2 (Table 1). The gap decreases for the bigger downstream model, but it is still there. The same trend can be observed for the SigLIP2-based models (Table 11). We argue this is due to the object-centric representations encoding the necessary information for the task more explicitly, making it easier for the smaller downstream model to extract the relevant information (Mamaghan et al., 2024).

**Expansion with Cross-Attention.** For the other direction of increasing the size of the object-centric representation, it is almost always worse for compositional generalization across all datasets and downstream models (Tables 1 and 11), both compared to the then same-sized original representation, e.g., DINOSAURv2 + CA versus DINOv2, and to the representation before increasing its size, e.g., DINOSAURv2 + CA versus DINOSAURv2.

## 4.3 THE EFFECT OF DOWNSTREAM COMPUTE

**Small Compute Budgets.** The COOD accuracies at different compute budgets and training difficulties of Super-CLEVR for the DINO-family are shown in Fig. 3. For small compute budgets – up to roughly four PFLOPs, the end of training for smaller image representations – and generalization

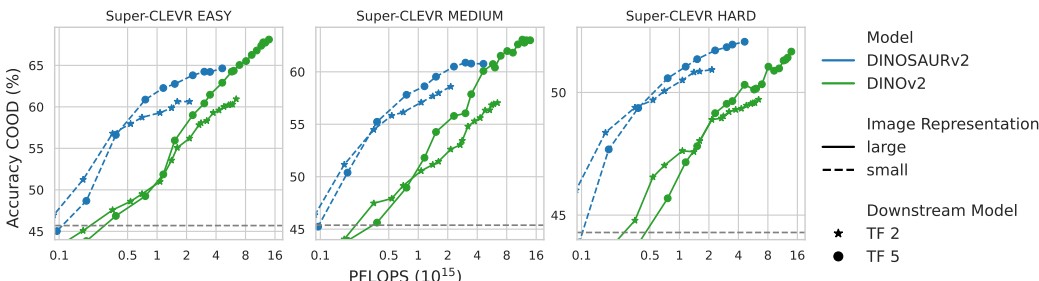

Figure 3: VQA COOD accuracy for training on Super-CLEVR *"easy"* (left), *"medium"* (middle), and *"hard"* (right) at different compute budgets (log FLOPs) with DINOv2-based models and question-only baseline (dashed grey). Object-centric representations slightly surpassed on easier generalization tasks (left) but are not matched on harder compositional out-of-distribution tasks (right) by dense representations, even at $3 \times$ the compute.

difficulties, Slot Attention-based object-centric representations consistently outperform all other representations.

**Easy Generalizations.** To surpass the COOD performance of DINOSAURv2 for easier generalization tasks, the non-object-centric counterpart, DINOv2, requires substantially more downstream compute. Even then, the eventual final accuracy gain is modest ($\leq 3.5\%$). For Super-CLEVR *"easy"* in Fig. 3 (left), DINOv2 reaches the best generalization accuracy of DINOSAURv2 with $1.5 \times$ the compute and improves $+3.5\%$ at the end after consuming $3 \times$ the computational resources. The same observations can be made for the SigLIP2-based models (Table 11).

**Hard Generalizations.** For the settings with the hardest compositional generalization, for example, Super-CLEVR *"hard"* (Fig. 3 right), small object-centric representations consistently match or outperform their non-object-centric counterparts within the same backbone family at any given compute budget. Even when granting original vision encoders up to $3 \times$ the compute, they often fail to surpass the object-centric representations (Table 11).

**Small Downstream Model.** Considering the performance of representations across downstream models, employing a smaller downstream model for the best COOD performance is justified only under very constrained compute budgets (for example, below 0.5 PFLOPs in Fig. 3). Under these limited compute conditions, DINOSAURv2's or SigLIPSAUR2's small image representation paired with the small downstream model consistently outperforms all alternatives.

### 4.4 THE EFFECT OF SAMPLE SIZE

**Varying Sample Size.** To examine how the amount of training data and its diversity affect compositional generalization, we train both the original vision encoders and their object-centric counterparts on subsets of all datasets. Specifically, we vary the sample size, defined as the number of training images, from $2^{10}$ (1024) up to $2^{15}$ (32768), each paired with corresponding question–answer pairs, and compare results to training on the full set of 40k images.

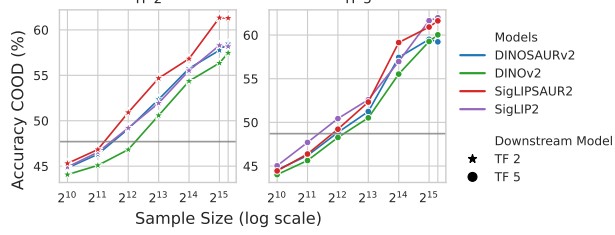

Figure 4: Object-centric models are more sample efficient, especially at lower sample sizes or with smaller downstream models. Compositional generalization of models trained on different subsets of the full data for MOVi-C *"easy"* for the small (left: TF 2) and bigger downstream model (right: TF 5). Question-Only baseline trained on full data in grey.

**Sample Size vs Highest Diversity.** When training on subsets of the dataset with the highest diversity, here shown for MOVi-C *"easy"* as an example in Fig. 4, the object-centric models consistently achieve better compositional generalization than their non-object-centric counterparts across all sample sizes when paired with the small downstream model (Fig. 4 left). In contrast, with more computational resources (larger downstream model, right), non-object-centric

representations can match or slightly surpass object-centric representations at larger sample sizes, here only at the largest sample size of the full dataset (40k). This indicates that object-centric models are more sample-efficient, likely because their smaller representations explicitly decompose the visual content of objects into different tokens, i.e., slots.

**Sample Size vs Diversity.** Comparing the effect of data diversity, the number of unique objects we see during training in *"easy"* – *"hard"*, across sample sizes: models trained on more diverse data almost always generalize better compositionally, especially at higher sample sizes. For example, for MOVi-C in Fig. 5, a higher diversity is always better for generalization for both DINOv2 and DINOSAURv2.

For all models and datasets, there always exists a breakpoint where increasing (i.e. doubling) the number of samples improves the compositional generalization ability more for higher compared to lower diversities. Concretely, for MOVi-C in Fig. 5, both models struggle to clear the question-only baseline (trained on the full data) for lower sample sizes, then both models across all three diversities behave pretty similarly until $2^{13}$ (8k) samples when doubling the number of samples again to $2^{14}$ brings a bigger improvement for higher diversities compared to lower ones for both models, where the improvement is bigger for the object-centric representation of DINOSAURv2. The sample size for this breakpoint can depend on the dataset, model, and diversity, but the overall trend is consistent. Finally, the accuracy for lower diversities and higher sample sizes can plateau, or even go down, depending on the dataset, model, and downstream model. Lastly, nearly all models match or surpass their compositional performance of the second-most diverse dataset (*"medium"*, full sample) with a lot fewer samples from the most diverse dataset (*"easy"*), here for MOVi-C with only $2^{14}$ images (around 40% of the full data), indicating that diversity is more important than sample size for generalization.

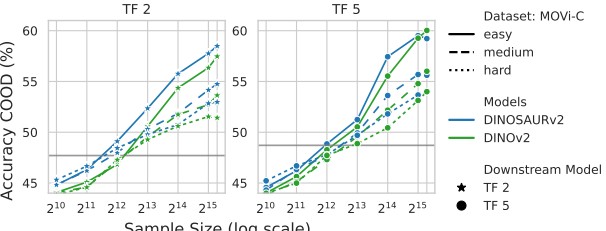

Figure 5: Object-centric models generalize better compositionally at lower diversities across almost all sample sizes. Compositional generalization of DINOv2 and DINOSAURv2 trained on different sample sizes of the full data for all diversities of MOVi-C *"easy"*– *"hard"* for the small (left: TF 2) and bigger downstream model (right: TF 5). Question-only baseline trained on full data in grey.

**OC vs Dense.** Focusing now on the difference between models, the object-centric representation is better at generalizing than its dense counterpart for all diversities across all sample sizes for the small downstream model[4], sometimes even comparing across diversities (cite fig). For the bigger downstream model, the dense representations only surpass the OC model at higher diversities and sample sizes. For MOVi-C in Fig. 5 (right), DINOv2 only surpasses DINOSAURv2 on *"easy"* and *"medium"* for the highest sample size. If we restrict one or both of diversity and sample size enough, the object-centric representation generalizes equally well or better.

## 5 CONCLUSION

In this work, we systematically evaluated the compositional generalization capabilities of object-centric representations in fully controlled and visually rich settings. By introducing a benchmark based on the CLEVRTex, Super-CLEVR, and MOVi-C datasets, we demonstrated that object-centric models, specifically DINOSAURv2 and SigLIPSAUR2, exhibit superior compositional generalization compared to their non-object-centric alternatives, DINOv2 and SigLIP2, while requiring significantly less compute.

These findings reinforce the potential of object-centric approaches for tasks requiring systematic compositional reasoning and highlight the need for further exploration into their applications beyond synthetic benchmarks. Future work may extend this by investigating the effectiveness of object-centric learning in real-world scenarios, incorporating more diverse datasets, and optimizing architectural choices to enhance performance across a broader range of vision tasks.

---

[4]We have observed this previously for all sample sizes of the *"easy"* versions.

## REPRODUCIBILITY STATEMENT

All the required information to reproduce the results is provided in the main text and the appendix. Specifically, model architectures, training and evaluation procedures, and hyperparameters are summarized in §3.2 and Appendix B. We use publicly available implementations of all models, which are properly cited in the paper. Datasets are generated from publicly released codebases, and the details of the data generation pipeline are documented in §3.1 and Appendix A.

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

## THE USE OF LARGE LANGUAGE MODELS (LLMS)

We used large language models solely as a writing assistant for language polishing (grammar, wording, and clarity). All scientific contributions, including problem formulation, dataset generation, experiments, and analyses, were produced by the authors.

## A    DATA GENERATION

The used attributes for image generation are in Table 2 for CLEVRTex, Table 3 for Super-CLEVR and Table 4 for MOVi-C, with some example images with question–answer pairs in Fig. 6, Fig. 7 and Fig. 8, respectively. For each base dataset, we define an object as the combination of all its attributes, and create three training datasets, which we label as *"easy"* (containing 80% of all possible objects), *"medium"* (40%), and *"hard"* (20%). We reserve the holdout 20% of object–property combinations for testing compositional generalization. During initial investigations, we found that for Super-CLEVR the generalization problem at these proportions is not "hard" enough yet, i.e., generalization behaves as in-distribution. This is likely due to the many factors of Super-CLEVR, of which not all are equally important for answering questions. Therefore, for Super-CLEVR only, we reduced the proportions to 10% (*"easy"*), 5% (*"medium"*) and 1% (*"hard"*).

Table 2: Attributes for the image and question generation for CLEVRTex.

| Shape (8) | Size (3) | Material (8) |
|---|---|---|
| cube | small | green tiled |
| cylinder | medium | blue denim |
| monkey head | large | red fabric |
| icosahedron | | green forest |
| teapot | | red leather |
| sphere | | rocky gravel |
| cone | | rusty metal |
| torus | | white sandstone |

Table 3: Attributes for the image and question generation for Super-CLEVR.

| Shape (21) | Size (2) | Material (2) | Color (8) | Texture (4) |
|---|---|---|---|---|
| suv | small | rubber | gray | none |
| wagon | large | metal | red | checkered |
| minivan | | | blue | stripped |
| sedan | | | green | dotted |
| truck | | | brown | |
| articulated bus | | | purple | |
| regular bus | | | cyan | |
| double bus | | | yellow | |
| school bus | | | | |
| chopper | | | | |
| dirtbike | | | | |
| scooter | | | | |
| cruiser | | | | |
| jet | | | | |
| fighter | | | | |
| biplane | | | | |
| airliner | | | | |
| road bike | | | | |
| utility bike | | | | |
| mountain bike | | | | |
| tandem bike | | | | |

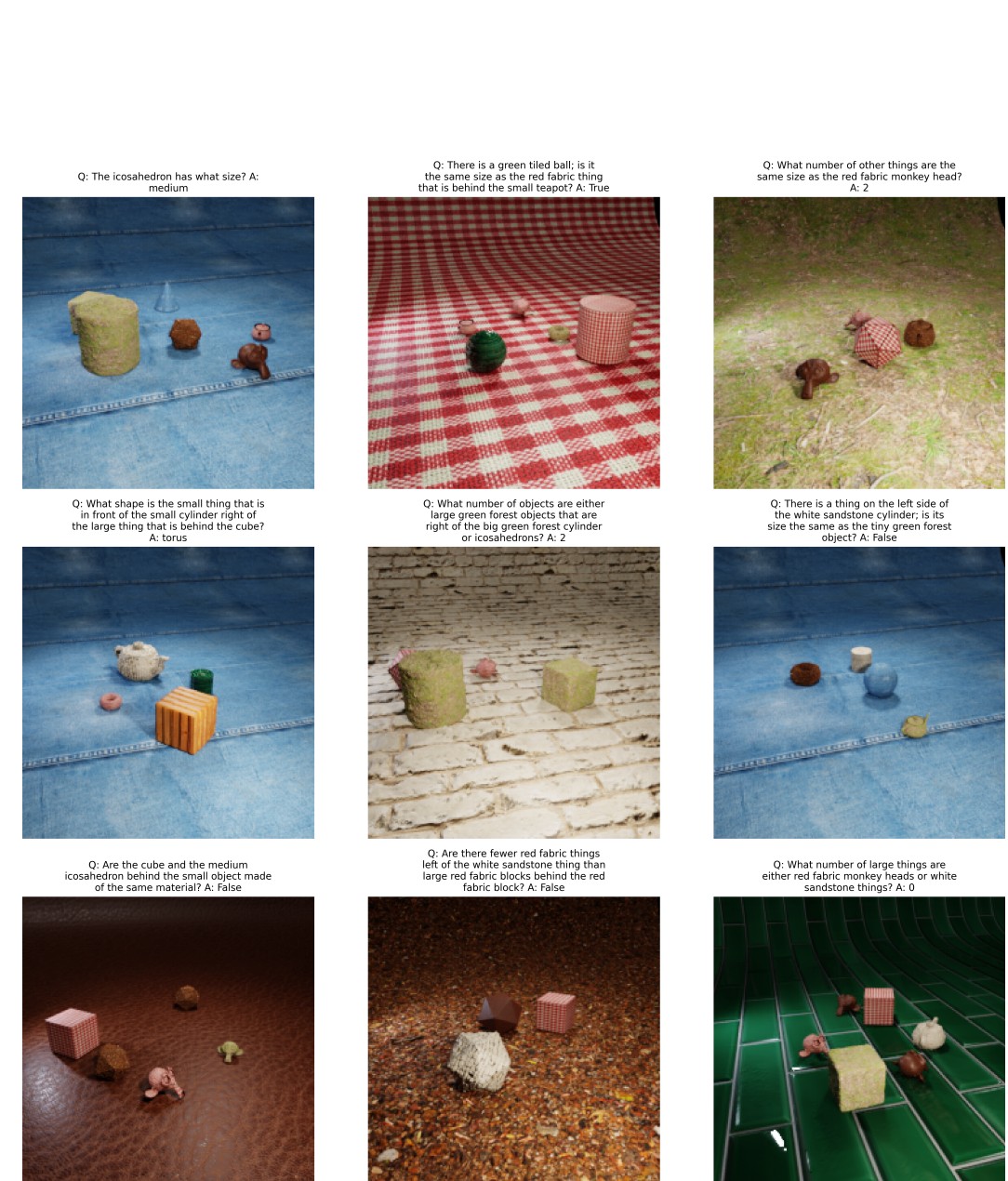

Figure 6: Dataset examples with question–answer pairs for CLEVRTex.

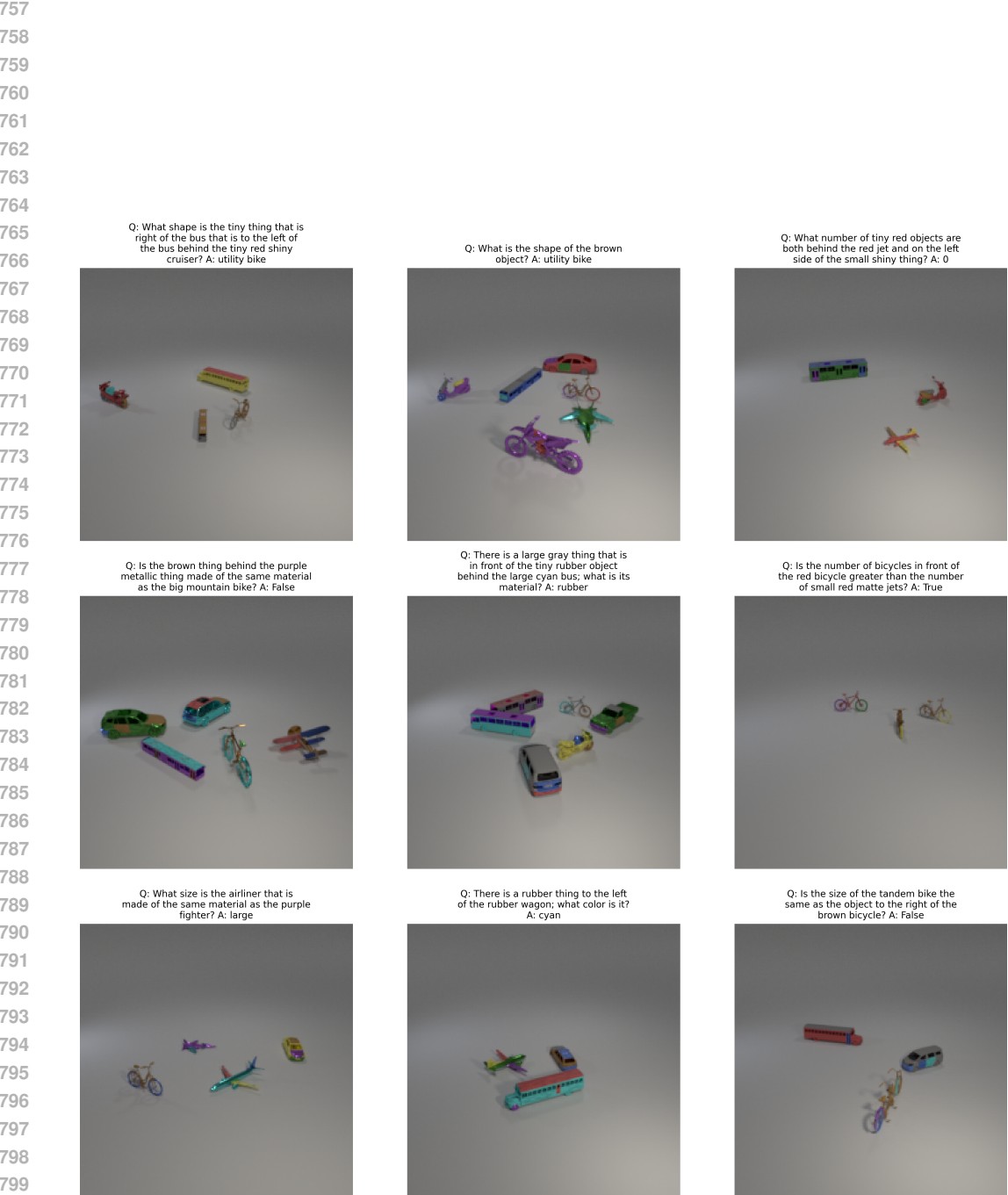

Figure 7: Dataset examples with question–answer pairs for Super-CLEVR.

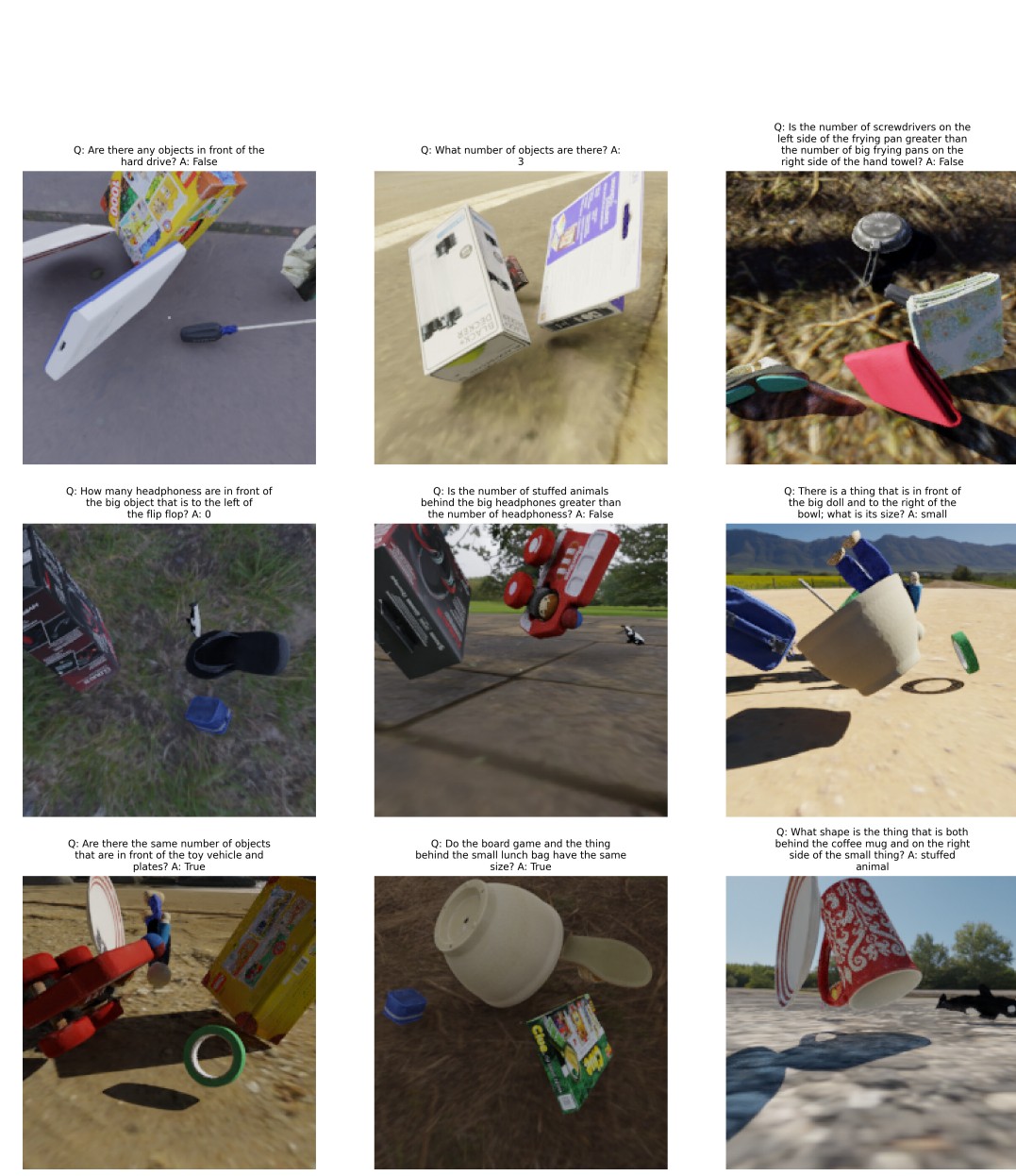

Figure 8: Dataset examples with question–answer pairs for MOVi-C.

Table 4: Attributes for the image and question generation for MOVi-C.

| **Shape** (36) | **Size** (3) |
| --- | --- |
| athletic shoe | small |
| boot | medium |
| sandal | large |
| flip flop | |
| ballet flat | |
| action figure | |
| stuffed animal | |
| board game | |
| puzzle toy | |
| construction toy | |
| toy vehicle | |
| musical toy | |
| doll | |
| game console | |
| ink cartridge | |
| computer mouse | |
| keyboard | |
| headphones | |
| router | |
| hard drive | |
| tablet | |
| coffee mug | |
| bowl | |
| plate | |
| hand towel | |
| screwdriver | |
| scissors | |
| tape roll | |
| plant pot | |
| lunch bag | |
| dish drying mat | |
| mixing bowl | |
| frying pan | |
| blender | |
| toaster | |
| coffee maker | |

## B  MODELS

### B.1  DINOSAURv2 AND SIGLIPSAUR2

The hyperparameters for DINOSAURv2 and SigLIPSAUR2 for all datasets can be found in Table 5.

### B.2  K-MEANS

For extracting an image representation via k-means from the pretrained vision encoders, we follow Baldassarre et al. (2024). Concretely, we concatenate the global image representation $g$ (from the CLS token) with the set of centroids derived from performing K-Means on the patch tokens for different numbers of clusters. After choosing the maximum number of clusters $k_{max}$, as a power of two, we use the standard sklearn[5] implementation for $k \in \{1, 2, \ldots, 2^{log_2(k_{max})}\}$. In contrast to Baldassarre et al. (2024), who mostly use a smaller $k_{max} = 8$ or $16$ for their more "global" tasks, e.g, scene classification and action recognition in videos, our VQA tasks require a more fine-grained

---

[5]https://scikit-learn.org/

spatial understanding and knowledge of visual details to distinguish between spatial relationships (e.g., left or right) and different objects (e.g., mountain or road bike). In preliminary experiments, we tried using $k_{max} \in \{8, 16, 32, 64\}$, but image representations with $k_{max} \leq 32$ performed quite bad, especially on CLEVRTex. In order to achieve reasonable performance while still resulting in a smaller image representation, therefore requiring less compute, we chose $k_{max} = 64$ which results in a representation with $128 = 1\,(\text{g}) + 1\,(\text{k=1}) + 1\,(\text{k=2}) + \ldots + 64\,(\text{k=64})$ tokens and feature dimension corresponding to the original vision encoder. For compute comparisons, refer to Appendix C.

Table 5: Hyperparameters of DINOSAURv2 and SigLIPSAUR2.

| Hyperparameter | | DINOSAURv2 | SigLIPSAUR2 |
|---|---|---|---|
| Training Steps | | 300k | 300k |
| Batch Size | | 128 | 128 |
| LR Warmup Steps | | 10k | 10k |
| Peak LR | | 0.0003 | 0.0002 |
| LR Schedule | | Cosine | Exp. Decay |
| Exp. Decay Half-Life | | - | 100k |
| Cosine T-Max | | 300k | - |
| Feature Extractor | | DINOv2_S | SigLIP2_B |
| Patch Size | | 14 | 16 |
| Feature Dim. | | 384 | 768 |
| Gradient Norm Clipping | | 0.1 | 0.1 |
| Image Size | | 224 | 224 |
| Cropping Strategy | | Full | Full |
| Image Tokens | | 256 | 196 |
| Decoder | Type | MLP | MLP |
| | Layers | 4 | 4 |
| | MLP Hidden Dim. | 2048 | 2048 |
| Slot Attention | Iterations | 3 | 3 |
| | Number of Slots | 7 | 7 |
| | Slot Dim. | 256 | 256 |
| | MLP Hidden Dim. | 1024 | 1024 |

### B.3 DOWNSTREAM VQA MODEL

**Architecture** We adopt a transformer-based architecture for VQA, following Mamaghan et al. (2024). We first project both image and text representations via separate linear layers (output size 126) with a dropout of 0.1, and augment them with a two-dimensional one-hot vector to indicate whether they originate from image features or text embeddings. We then add a sinusoidal positional encoding to the text embeddings. To perform classification, we use a trainable $\text{CLS} \in \mathbb{R}^{128}$ vector. We concatenate the image and text representations (plus the CLS token) and pass them through a transformer encoder with $d_{model} = 128$ and a hidden dimension of 128. The transformed CLS token is fed into a two-layer MLP (hidden dimension 128) with layer normalization, a dropout rate of 0.1, and a ReLU activation between layers. This MLP outputs a probability distribution over all possible answers.

**Training** For all CLEVRTex, Super-CLEVR and MOVi-C variants, we train the downstream models with a batch size of 128, a learning rate of 0.0001, and a cross-entropy loss for steps defined in section Appendix C. We use downstream model variants where we vary the number of layers of the transformer encoder, either 2 or 5 layers with 64 heads. We tried changing the learning rate and schedule, including linear warm-up and/or different learning rate schedules, e.g., cosine, but all of them resulted in worse performance compared to the above setting.

## C COMPUTE

The base models DINOv2, SigLIP2, and DINOSAURv2/SigLIPSAUR2 produce for all datasets here, i.e., with an image size of 224, representations of shape $[256, 384]$, $[196, 768]$, and $[7, 256]$,

respectively. This results in a huge compute mismatch for the downstream model[6] in Fig. 9, where, for example, the FLOPs for the downstream model with the DINOv2 image representation are roughly four times that of DINOSAURv2 for both transformer sizes. To remedy that, we use a single cross-attention (CA) layer with four heads right after the vision encoder to map from the large, i.e., the shape for DINOv2, to the small image representation or vice-versa. We additionally considered mapping from the input to the same output size and using a different number of layers or heads for CA, but none of this resulted in consistent improvements. In order to choose the number of steps to train each model, we train the downstream model with the image representation that needs the most compute, i.e., for DINOv2 see Fig. 9, until convergence for 600k steps (same as Mamaghan et al. (2024)). Then, for each training checkpoint of this model, the union of every 50k steps and a power of two series, we calculate the corresponding number of steps for all other models, depending on their compute ratios. As this results in a very high number of steps for some of the models with smaller image representations, e.g., DINOSAURv2, we choose a reasonable (fixed) number of max steps for which models are already converged or learning slowed down a lot. For the number of steps used consistently for all models shown here, see Table 6.

Table 6: Number of steps for the two downstream models for all image representations.

| Model Name | Image Repr. Size | Number of steps | |
| --- | --- | --- | --- |
| | | TF 2 | TF 5 |
| SigLIP2 / SigLIPSAUR2 + CA | [196,768] | 641123 | 688733 |
| DINOv2 / DINOSAURv2 + CA | [256,384] | 600000 | 600000 |
| SigLIP2 + K-Means | [128,768] | 718928 | 617749 |
| DINOv2 + K-Means | [128,384] | 643602 | 652251 |
| SigLIPSAUR2 / DINOSAURv2 / | [7,256] | 762329 | 809568 |
| SigLIP2 + CA / DINOv2 + CA / | | | |
| SigLIP2 + K-Means (7) / DINOv2 + K-Means (7) | | | |

# D    ADDITIONAL COMPARISONS

Table 7: VQA Accuracy in-distribution for all image representations and the small downstream model (TF 2).

| TF 2 | CLEVRTex | | | Super-CLEVR | | | MOVi-C | | |
| --- | --- | --- | --- | --- | --- | --- | --- | --- | --- |
| | E | M | H | E | M | H | E | M | H |
| Question-Only Baseline | 46.3 | 48.1 | 53.0 | 49.0 | 50.7 | 59.5 | 54.2 | 55.7 | 57.5 |
| DINOv2 | 73.9 | 76.6 | 82.6 | 65.9 | 68.2 | 81.2 | 72.8 | 75.0 | 78.0 |
| DINOv2 + CA | 73.1 | 71.2 | 76.3 | 65.5 | 66.7 | 79.4 | 69.6 | 72.4 | 74.0 |
| DINOv2 + KMeans (7) | 59.2 | 62.2 | 68.6 | 55.5 | 58.0 | 70.9 | 61.7 | 64.2 | 66.5 |
| DINOv2 + KMeans | 73.1 | 76.3 | 80.6 | 65.1 | 67.7 | 80.5 | 72.4 | 74.8 | 77.3 |
| DINOSAURv2 | 79.0 | 82.9 | 85.4 | 67.6 | 70.2 | 81.8 | 74.3 | 76.7 | 78.7 |
| DINOSAURv2 + CA | 73.8 | 80.4 | 81.0 | 63.0 | 65.6 | 78.4 | 68.8 | 72.3 | 74.5 |
| SigLIP2 | 77.8 | 79.6 | 83.6 | 68.1 | 69.3 | 82.5 | 73.1 | 76.4 | 78.7 |
| SigLIP2 + CA | 69.4 | 73.8 | 78.6 | 65.4 | 67.0 | 80.3 | 69.1 | 72.5 | 75.3 |
| SigLIP2 + KMeans (7) | 60.4 | 63.1 | 70.0 | 57.2 | 60.1 | 72.6 | 62.8 | 65.3 | 68.2 |
| SigLIP2 + KMeans | 78.3 | 78.6 | 81.8 | 66.9 | 70.1 | 81.8 | 73.5 | 75.8 | 78.5 |
| SigLIPSAUR2 | 84.7 | 85.2 | 87.1 | 71.0 | 73.4 | 84.1 | 76.8 | 79.0 | 81.0 |
| SigLIPSAUR2 + CA | 74.8 | 77.5 | 82.0 | 66.6 | 68.4 | 80.5 | 71.9 | 74.2 | 76.3 |
| GT Oracle | 95.3 | 95.4 | 96.1 | 91.3 | 91.6 | 93.5 | 95.0 | 95.7 | 95.7 |

---

[6] https://github.com/facebookresearch/fvcore

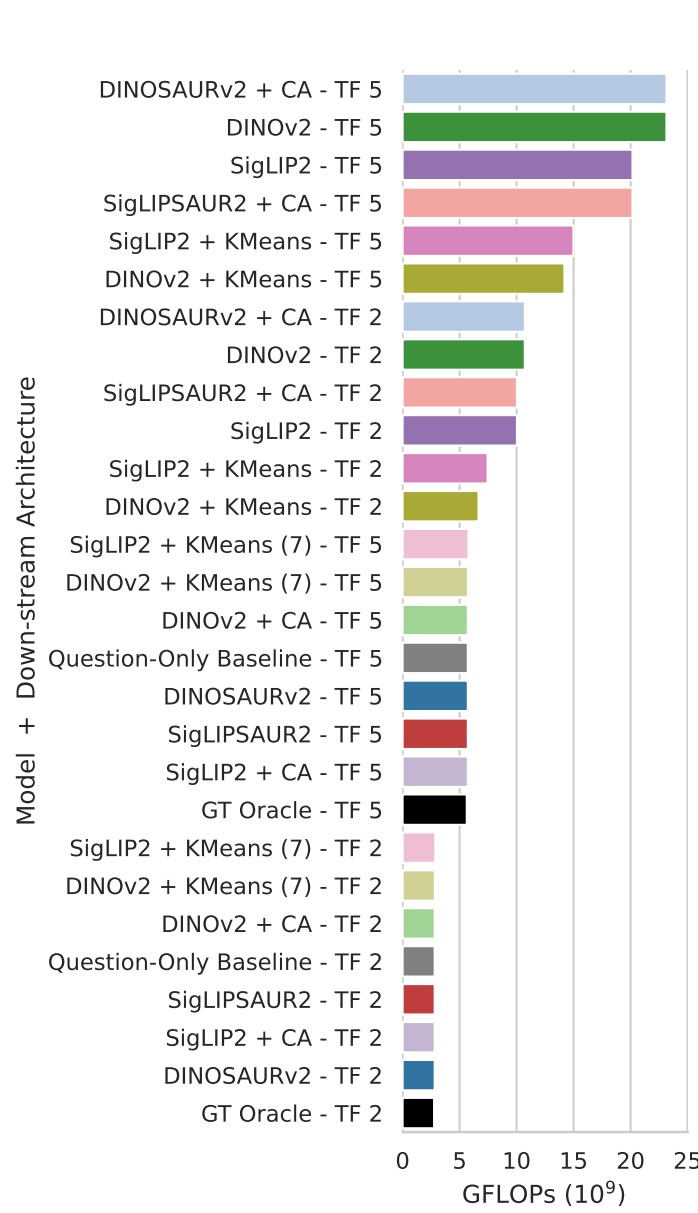

Figure 9: GFLOPs for one step of the downstream model for image representations with both the smaller (TF 2) and bigger transformer downstream model (TF 5), ignoring the compute needed for resizing with the cross-attention layer.

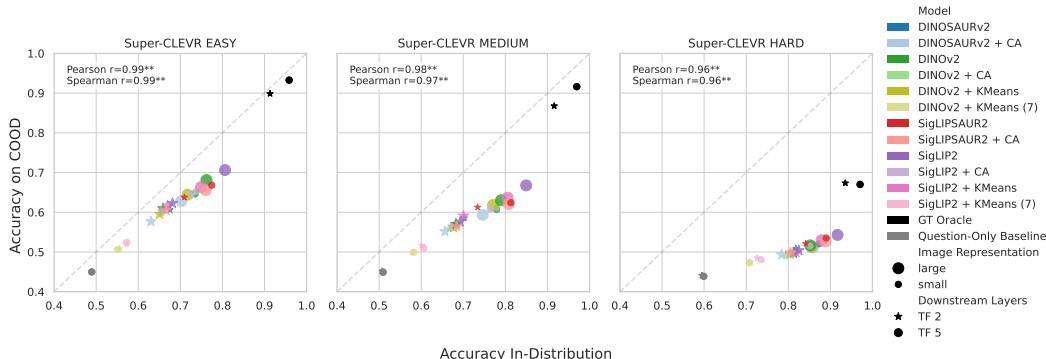

Figure 10: VQA in-distribution and compositional out-of-distribution accuracy are very strongly correlated (highly significant: p-value < .01). Performances for Super-CLEVR *"easy"*, *"medium"*, and *"hard"* at the end of training with correlations and ground-truth oracle (upper right: black) and question-only baseline (lower left: grey).

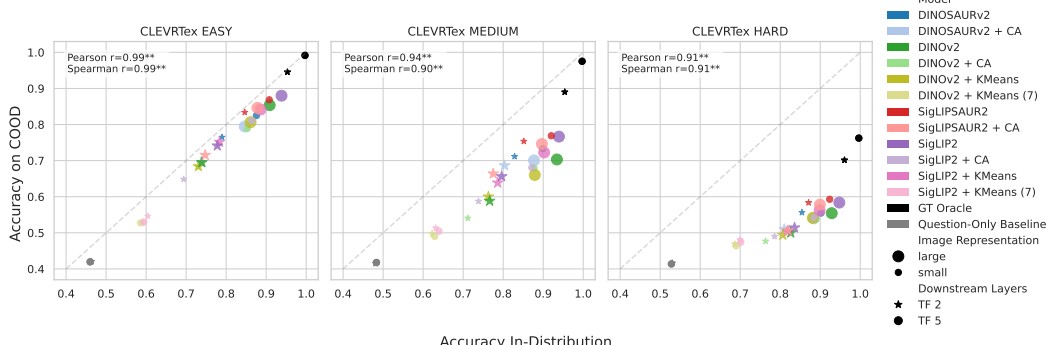

Figure 11: VQA in-distribution and compositional out-of-distribution accuracy are very strongly correlated (highly significant: p-value < .01). Performances for CLEVRTex *"easy"*, *"medium"*, and *"hard"* at the end of training with correlations and ground-truth oracle (upper right: black) and question-only baseline (lower left: grey).

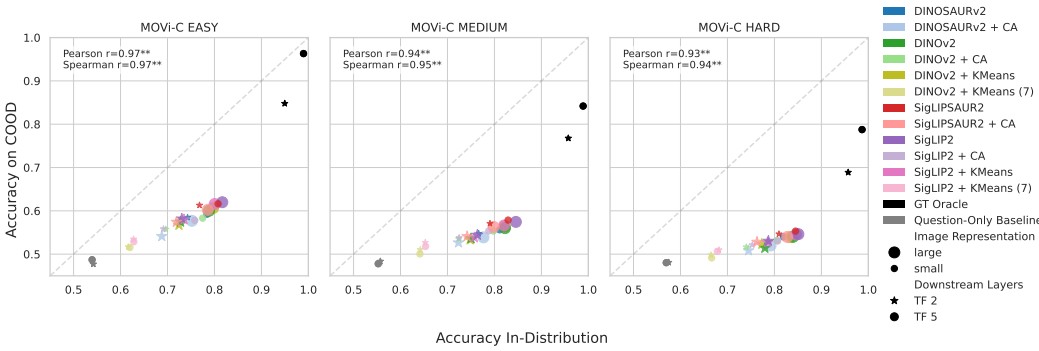

Figure 12: VQA in-distribution and compositional out-of-distribution accuracy are very strongly correlated (highly significant: p-value < .01). Performances for MOVi-C *"easy"*, *"medium"*, and *"hard"* at the end of training with correlations and ground-truth oracle (upper right: black) and question-only baseline (lower left: grey).

Table 8: VQA Accuracy compositional out-of-distribution for all image representations and the small downstream model (TF 2).

| TF 2 | CLEVRTex | | | Super-CLEVR | | | MOVi-C | | |
|---|---|---|---|---|---|---|---|---|---|
| | E | M | H | E | M | H | E | M | H |
| Question-Only Baseline | 42.1 | 41.5 | 41.6 | 45.0 | 45.2 | 44.1 | 47.7 | 48.4 | 48.1 |
| DINOv2 | 69.5 | 58.8 | 50.0 | 60.9 | 57.0 | 49.7 | 57.5 | 53.6 | 51.4 |
| DINOv2 + CA | 68.4 | 54.1 | 47.7 | 59.8 | 55.8 | 48.9 | 55.8 | 53.6 | 51.7 |
| DINOv2 + KMeans (7) | 53.0 | 49.8 | 46.9 | 50.8 | 49.9 | 47.5 | 51.7 | 51.0 | 49.8 |
| DINOv2 + KMeans | 68.4 | 60.0 | 49.4 | 59.5 | 56.2 | 49.6 | 56.7 | 53.4 | 52.4 |
| DINOSAURv2 | 76.5 | 71.2 | 55.6 | 60.6 | 58.6 | 50.9 | 58.5 | 54.7 | 53.0 |
| DINOSAURv2 + CA | 69.6 | 68.7 | 51.0 | 57.7 | 55.1 | 49.4 | 54.1 | 52.7 | 50.8 |
| SigLIP2 | 74.1 | 65.6 | 51.4 | 62.3 | 57.6 | 50.4 | 58.2 | 54.4 | 53.1 |
| SigLIP2 + CA | 64.8 | 58.8 | 49.0 | 60.4 | 56.5 | 49.5 | 55.8 | 53.8 | 52.1 |
| SigLIP2 + KMeans (7) | 54.6 | 51.4 | 48.1 | 52.0 | 51.6 | 48.5 | 53.5 | 52.7 | 51.0 |
| SigLIP2 + KMeans | 75.2 | 63.9 | 50.5 | 61.6 | 59.1 | 49.9 | 58.0 | 53.8 | 52.7 |
| SigLIPSAUR2 | 83.4 | 75.4 | 58.4 | 63.8 | 61.3 | 52.1 | 61.3 | 57.1 | 54.7 |
| SigLIPSAUR2 + CA | 71.5 | 66.4 | 50.7 | 60.5 | 56.4 | 49.9 | 57.4 | 54.1 | 52.8 |
| GT Oracle | 94.6 | 89.0 | 70.2 | 89.9 | 86.8 | 67.3 | 84.8 | 76.8 | 68.9 |

Table 9: VQA Accuracy in-distribution for all image representations and the big downstream model (TF 5).

| TF 5 | CLEVRTex | | | Super-CLEVR | | | MOVi-C | | |
|---|---|---|---|---|---|---|---|---|---|
| | E | M | H | E | M | H | E | M | H |
| Question-Only Baseline | 46.0 | 48.3 | 52.8 | 48.9 | 50.9 | 59.9 | 53.9 | 55.3 | 57.0 |
| DINOv2 | 90.9 | 93.4 | 92.9 | 76.2 | 79.0 | 85.2 | 78.8 | 82.2 | 84.0 |
| DINOv2 + CA | 85.3 | 87.7 | 88.9 | 73.7 | 77.9 | 85.2 | 77.5 | 79.8 | 80.6 |
| DINOv2 + KMeans (7) | 58.6 | 62.9 | 69.0 | 55.0 | 58.2 | 70.8 | 62.0 | 64.1 | 66.6 |
| DINOv2 + KMeans | 86.1 | 87.9 | 88.2 | 71.7 | 77.1 | 85.5 | 79.7 | 82.2 | 83.4 |
| DINOSAURv2 | 87.6 | 89.9 | 90.2 | 73.5 | 77.9 | 87.3 | 78.4 | 81.1 | 81.9 |
| DINOSAURv2 + CA | 84.7 | 87.7 | 88.4 | 70.3 | 74.6 | 85.9 | 75.2 | 77.7 | 79.2 |
| SigLIP2 | 93.8 | 93.9 | 94.8 | 80.6 | 84.9 | 91.7 | 81.7 | 84.6 | 85.1 |
| SigLIP2 + CA | 86.4 | 87.2 | 88.4 | 72.8 | 76.4 | 85.1 | 75.8 | 78.8 | 80.7 |
| SigLIP2 + KMeans (7) | 59.4 | 63.9 | 70.2 | 57.3 | 60.6 | 73.5 | 62.8 | 65.3 | 67.8 |
| SigLIP2 + KMeans | 88.6 | 90.2 | 89.9 | 74.9 | 80.5 | 87.9 | 80.0 | 82.1 | 84.6 |
| SigLIPSAUR2 | 90.8 | 92.0 | 92.4 | 77.5 | 81.3 | 89.0 | 80.8 | 82.9 | 84.5 |
| SigLIPSAUR2 + CA | 87.8 | 89.7 | 89.9 | 76.0 | 80.8 | 88.9 | 78.6 | 79.8 | 82.7 |
| GT Oracle | 99.7 | 99.7 | 99.6 | 95.8 | 97.0 | 97.1 | 99.0 | 98.9 | 98.7 |

Table 10: VQA Accuracy compositional out-of-distribution for all image representations and the big downstream model (TF 5).

| TF 5 | CLEVRTex | | | Super-CLEVR | | | MOVi-C | | |
|---|---|---|---|---|---|---|---|---|---|
| | E | M | H | E | M | H | E | M | H |
| Question-Only Baseline | 42.0 | 41.8 | 41.4 | 45.0 | 44.9 | 43.9 | 48.7 | 47.8 | 48.0 |
| DINOv2 | 85.4 | 70.3 | 55.4 | 68.1 | 63.0 | 51.7 | 60.0 | 56.0 | 54.0 |
| DINOv2 + CA | 78.9 | 67.9 | 53.8 | 65.2 | 61.1 | 50.8 | 58.3 | 55.3 | 53.2 |
| DINOv2 + KMeans (7) | 52.7 | 49.0 | 46.4 | 50.6 | 49.9 | 47.2 | 51.5 | 50.1 | 49.1 |
| DINOv2 + KMeans (128) | 80.6 | 66.0 | 54.1 | 64.4 | 61.8 | 51.3 | 60.6 | 56.5 | 53.9 |
| DINOSAURv2 | 82.5 | 73.3 | 55.5 | 64.6 | 60.8 | 52.1 | 59.2 | 55.6 | 54.0 |
| DINOSAURv2 + CA | 79.5 | 70.1 | 54.0 | 62.8 | 59.3 | 51.1 | 57.7 | 53.9 | 52.0 |
| SigLIP2 | 88.0 | 76.7 | 58.4 | 70.6 | 66.8 | 54.3 | 62.0 | 57.5 | 54.6 |
| SigLIP2 + CA | 81.1 | 68.1 | 54.2 | 64.7 | 60.9 | 51.5 | 57.8 | 55.4 | 53.0 |
| SigLIP2 + KMeans (7) | 53.0 | 50.5 | 47.4 | 52.4 | 50.9 | 48.1 | 52.9 | 51.8 | 50.6 |
| SigLIP2 + KMeans (128) | 84.2 | 72.3 | 56.2 | 66.3 | 63.7 | 52.9 | 61.6 | 56.6 | 54.5 |
| SigLIPSAUR2 | 86.9 | 76.9 | 59.3 | 66.8 | 62.4 | 53.5 | 61.6 | 57.9 | 55.3 |
| SigLIPSAUR2 + CA | 84.6 | 74.6 | 57.8 | 65.6 | 62.1 | 52.7 | 60.2 | 56.3 | 54.0 |
| GT Oracle | 99.2 | 97.5 | 76.2 | 93.3 | 91.6 | 67.0 | 96.3 | 84.2 | 78.8 |

Table 11: VQA accuracy (%) of both downstream models (TF 2 & 5) on the respective compositional generalization test sets for all models, trained on *"easy"* (E), *"medium"* (M), and *"hard"* (H) training sets. We compute deltas compared to the original pretrained vision encoder.

| TF 2 | CLEVRTex | | | Super-CLEVR | | | MOVi-C | | |
|---|---|---|---|---|---|---|---|---|---|
| | E | M | H | E | M | H | E | M | H |
| DINOv2 | 69.5 | 58.8 | 50.0 | 60.9 | 57.0 | 49.7 | 57.5 | 53.6 | 51.4 |
| DINOv2 + CA | -1.2 | -4.8 | -2.3 | -1.2 | -1.2 | -0.8 | -1.7 | -0.1 | 0.2 |
| DINOv2 + KMeans (7) | -16.5 | -9.1 | -3.1 | -10.1 | -7.1 | -2.2 | -5.8 | -2.6 | -1.6 |
| DINOv2 + KMeans | -1.1 | 1.2 | -0.6 | -1.4 | -0.8 | -0.1 | -0.7 | -0.3 | 0.9 |
| DINOSAURv2 | 7.0 | 12.3 | 5.6 | -0.3 | 1.6 | 1.2 | 1.0 | 1.1 | 1.6 |
| DINOSAURv2 + CA | 0.1 | 9.8 | 1.0 | -3.3 | -1.9 | -0.4 | -3.3 | -1.0 | -0.6 |
| SigLIP2 | 74.1 | 65.6 | 51.4 | 62.3 | 57.6 | 50.4 | 58.2 | 54.4 | 53.1 |
| SigLIP2 + CA | -9.3 | -6.9 | -2.4 | -1.9 | -1.0 | -0.9 | -2.3 | -0.6 | -0.9 |
| SigLIP2 + KMeans (7) | -19.5 | -14.3 | -3.3 | -10.3 | -6.0 | -1.9 | -4.7 | -1.7 | -2.1 |
| SigLIP2 + KMeans | 1.0 | -1.8 | -1.0 | -0.6 | 1.5 | -0.5 | -0.2 | -0.6 | -0.3 |
| SigLIPSAUR2 | 9.2 | 9.7 | 6.9 | 1.5 | 3.7 | 1.8 | 3.1 | 2.6 | 1.6 |
| SigLIPSAUR2 + CA | -2.6 | 0.8 | -0.8 | -1.8 | -1.2 | -0.5 | -0.7 | -0.3 | -0.2 |
| **TF 5** | E | M | H | E | M | H | E | M | H |
| DINOv2 | 85.4 | 70.3 | 55.4 | 68.1 | 63.0 | 51.7 | 60.0 | 56.0 | 54.0 |
| DINOv2 + CA | -6.5 | -2.4 | -1.7 | -2.9 | -1.9 | -0.9 | -1.7 | -0.7 | -0.8 |
| DINOv2 + KMeans (7) | -32.6 | -21.3 | -9.1 | -17.5 | -13.1 | -4.5 | -8.5 | -6.0 | -4.9 |
| DINOv2 + KMeans | -4.7 | -4.3 | -1.3 | -3.7 | -1.2 | -0.4 | 0.6 | 0.5 | -0.1 |
| DINOSAURv2 | -2.9 | 3.0 | 0.1 | -3.5 | -2.2 | 0.4 | -0.8 | -0.4 | -0.0 |
| DINOSAURv2 + CA | -5.9 | -0.3 | -1.4 | -5.3 | -3.7 | -0.6 | -2.3 | -2.1 | -2.0 |
| SigLIP2 | 88.0 | 76.7 | 58.4 | 70.6 | 66.8 | 54.3 | 62.0 | 57.5 | 54.6 |
| SigLIP2 + CA | -6.9 | -8.6 | -4.2 | -5.9 | -5.9 | -2.8 | -4.2 | -2.1 | -1.6 |
| SigLIP2 + KMeans (7) | -35.0 | -26.2 | -11.0 | -18.2 | -15.8 | -6.2 | -9.1 | -5.7 | -4.0 |
| SigLIP2 + KMeans | -3.8 | -4.4 | -2.1 | -4.3 | -3.1 | -1.4 | -0.4 | -0.8 | -0.1 |
| SigLIPSAUR2 | -1.1 | 0.2 | 0.9 | -3.8 | -4.4 | -0.8 | -0.4 | 0.4 | 0.7 |
| SigLIPSAUR2 + CA | -3.4 | -2.0 | -0.6 | -5.0 | -4.7 | -1.6 | -1.8 | -1.2 | -0.7 |

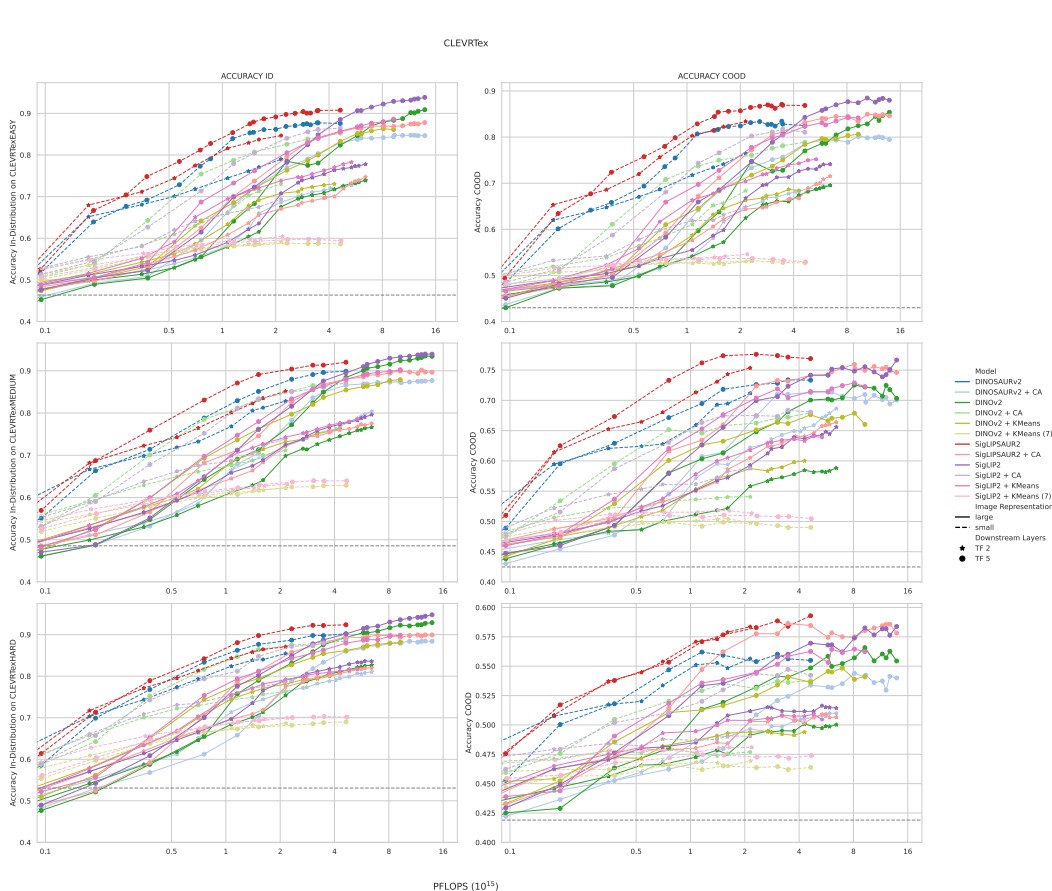

Figure 13: VQA in-distribution and compositional out-of-distribution accuracy for all CLEVRTex dataset variants with question-only baseline (lower: dashed grey).

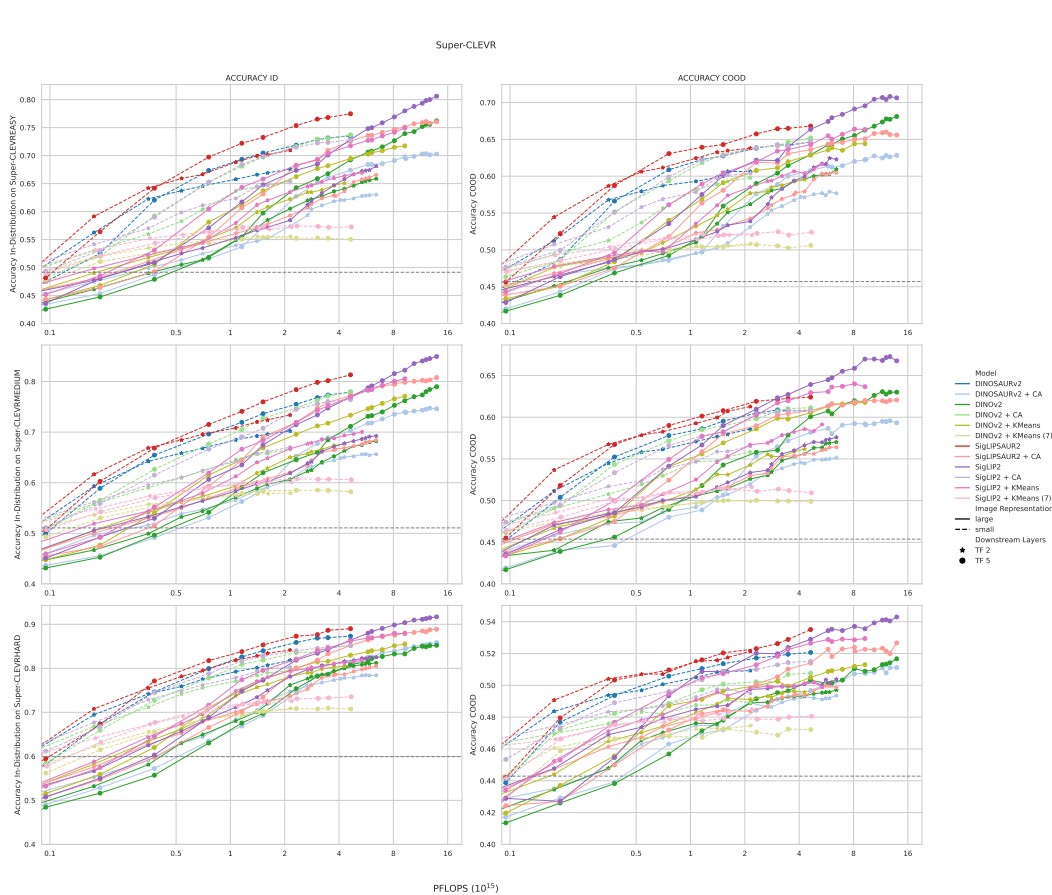

Figure 14: VQA in-distribution and compositional out-of-distribution accuracy for all Super-CLEVR dataset variants with question-only baseline (lower: dashed grey).

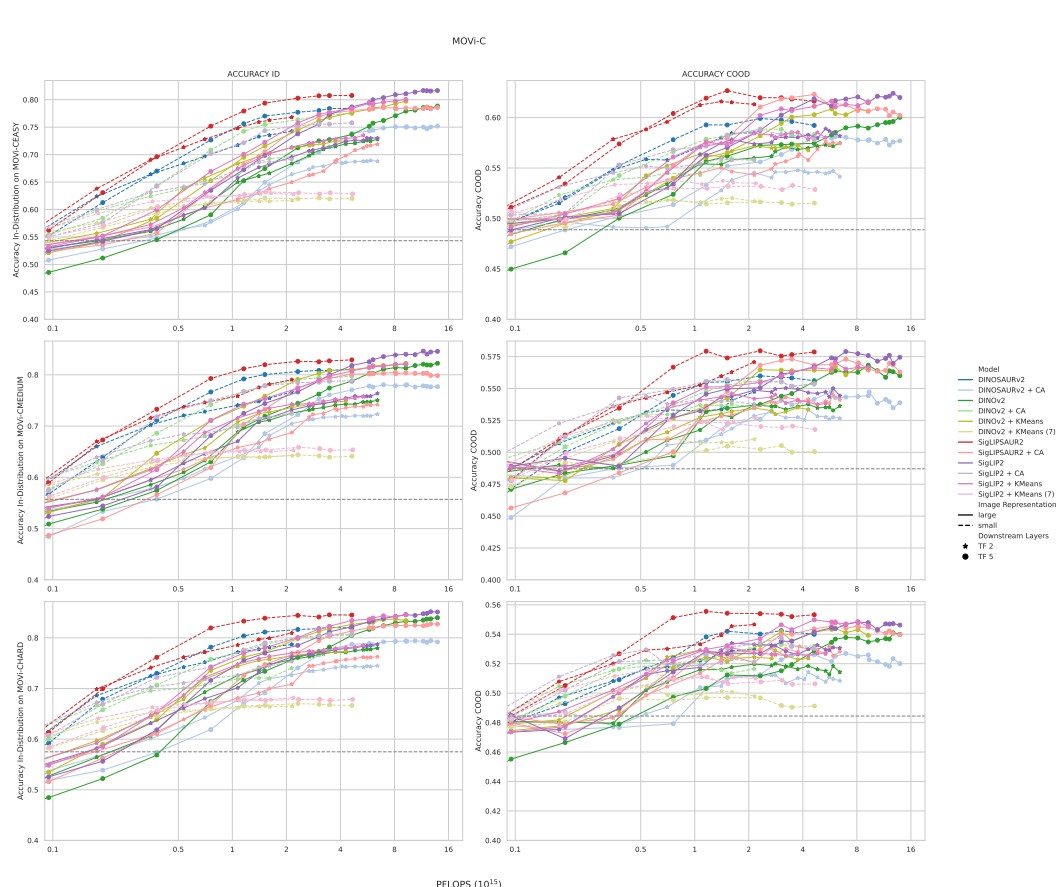

Figure 15: VQA in-distribution and compositional out-of-distribution accuracy for all MOVi-C dataset variants with question-only baseline (lower: dashed grey).

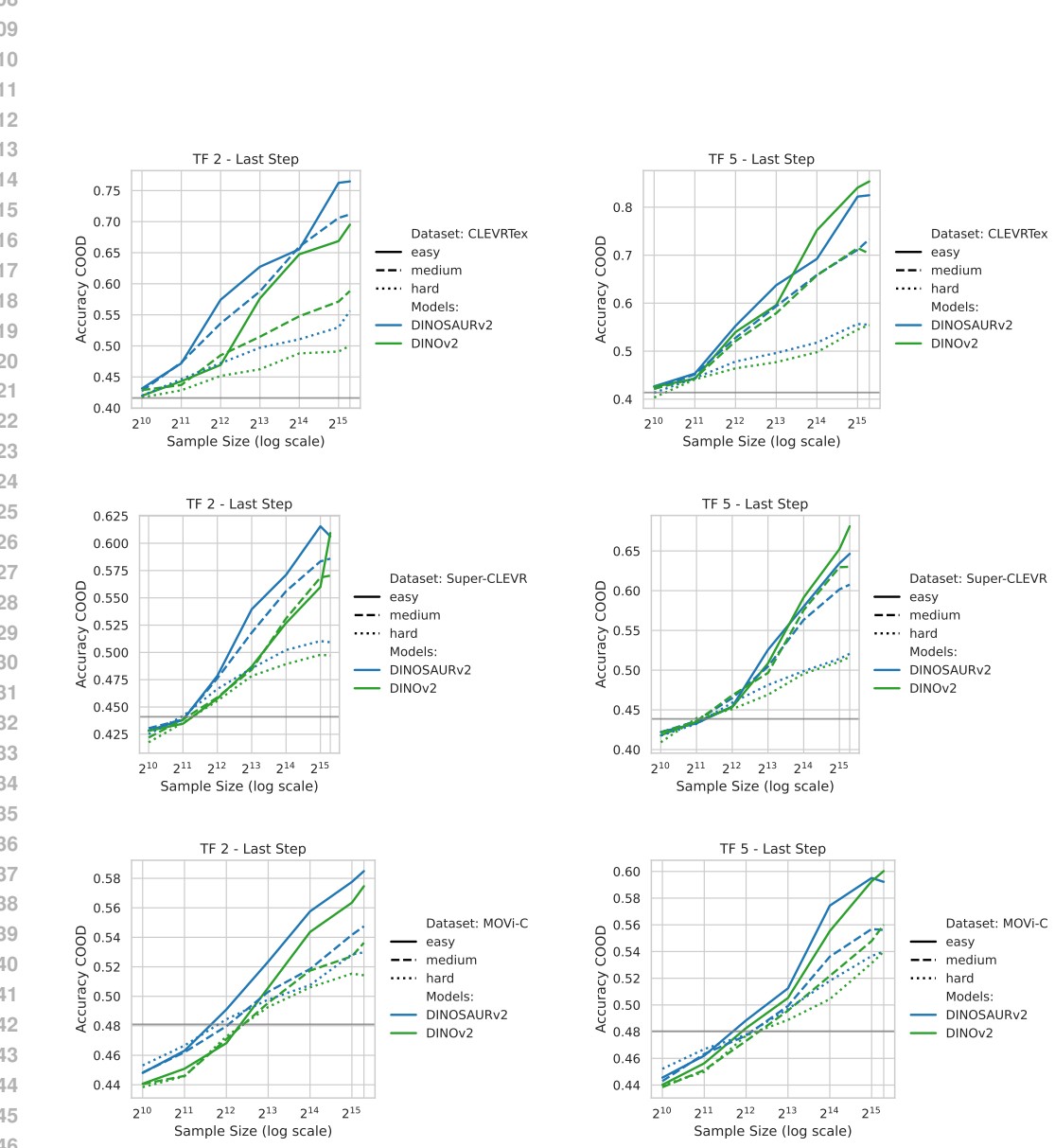

Figure 17: Compositional generalization of models trained on different subsets of the full data for CLEVRTex, Super-CLEVR and MOVi-C for DINOv2-based models.

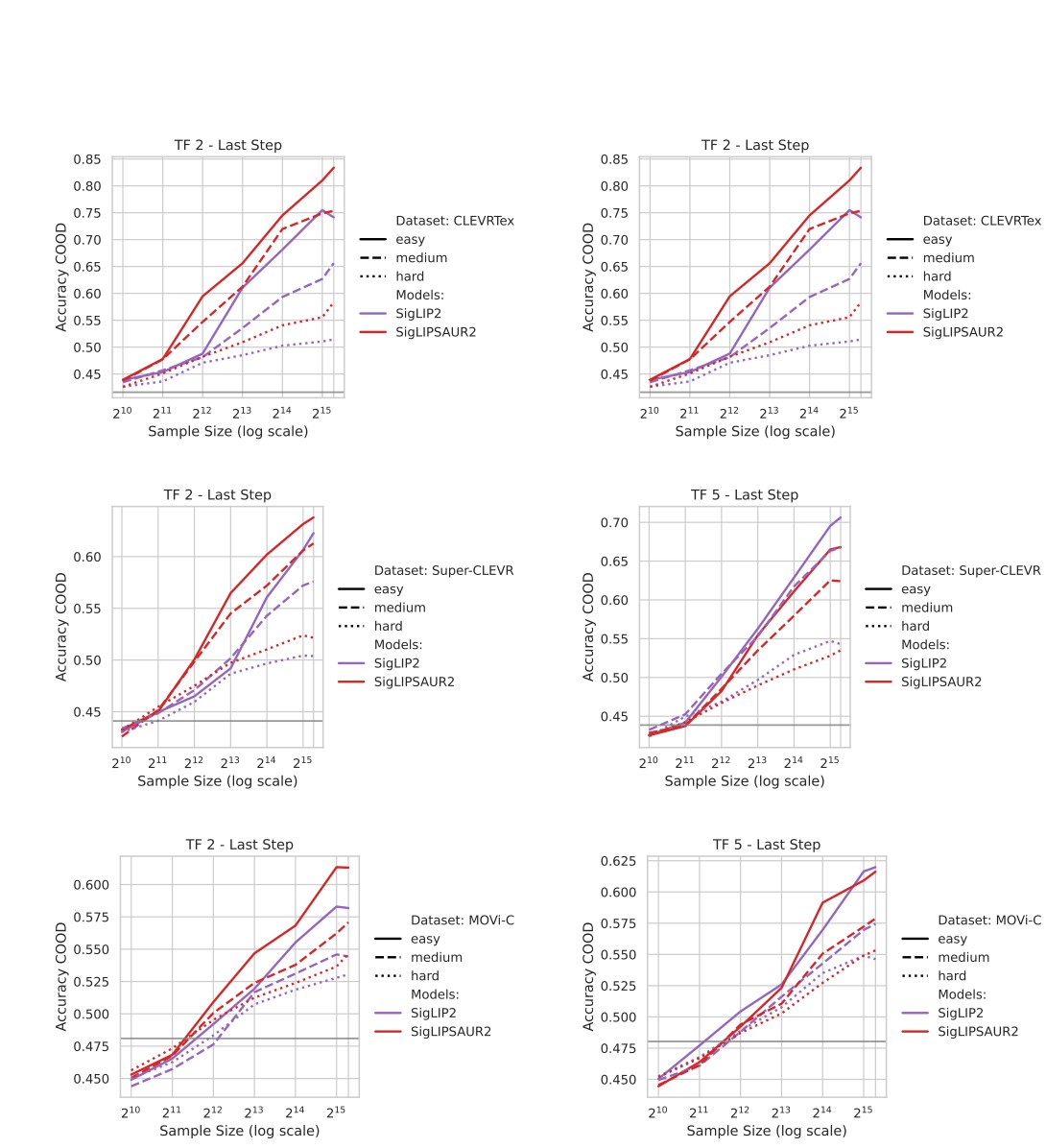

Figure 18: Compositional generalization of models trained on different subsets of the full data for CLEVRTex, Super-CLEVR and MOVi-C for SigLIP2-based models.

