# OpenReview forum: "Are Object-Centric Representations Better At Compositional Generalization?"
_ICLR.cc/2026/Conference — Submitted to ICLR 2026_

### Official Review · Reviewer_oVPf · 2025-10-30

**Soundness:** 2
**Presentation:** 3
**Contribution:** 3
**Rating:** 4
**Confidence:** 5

**Summary:**

This paper investigates whether object-centric (OC) representations improve compositional generalization—the ability to reason about novel combinations of familiar object properties—compared to conventional dense visual features. The authors construct a controlled Visual Question Answering (VQA) benchmark across three synthetic visual worlds (CLEVRTex, Super-CLEVR, MOVi-C), systematically varying training diversity, sample size, and downstream compute. Using pretrained foundation models (DINOv2, SigLIP2) and their object-centric counterparts (DINOSAURv2, SigLIPSAUR2), they perform fair, capacity-controlled comparisons. Experiments show that OC representations outperform dense ones in harder compositional generalization settings, are more sample-efficient, and require less compute to achieve comparable or better results. Dense models only surpass OC ones in easier tasks with large data and high compute budgets. Overall, the study provides the first systematic, quantitative evidence that object-centric representations lead to stronger compositional generalization, especially under limited data, diversity, or compute conditions.

**Strengths:**

- **Methodological Rigor:** The study offers a carefully controlled and systematic evaluation, isolating the effects of training diversity, compute, and sample size on compositional generalization.

- **Comprehensive Experiments:** The evaluation spans three synthetic visual worlds, multiple difficulty levels, and two major vision encoder families (DINOv2 and SigLIP2).

- **Benchmark Contribution:** The proposed VQA-based compositional generalization benchmark provides a valuable resource for future research on systematic generalization in vision.

**Weaknesses:**

- **The statement does not match the table: ** Regarding the dense features (such as the comparison between DINOv2 and DINOSAURv2), the paper states several main conclusions, including: OC performs better in complex compositional generalization scenarios; dense features only outperform OC in simple scenarios. However, according to the data in Table 1, the main advantage of DINOSAURv2 is reflected in TF 2+CLEVRTex and TF 5+CLEVRTex (medium difficulty). In other settings, compared with DINOv2, the advantage of DINOSAURv2 is not significant, and even its performance is lower. Moreover, as the generalization difficulty increases, DINOSAURv2 does not show a gradually increasing advantage (among the six comparisons in Table 1, in three of them, medium has a greater advantage than hard), which is inconsistent with the claim that "dense features only outperform OC in simple scenarios".

- **Scale of VQA model: ** Although the impact of network scale was discussed in the paper, the scales of the networks were not large: only 128 dimensions and 2/5 encoder layers were used. This setting is far from the scale of transformers typically used in VQA tasks. For instance, in some classic works on OCL+VQA, MDETR [1] uses 6 encoder + 6 decoder layers with 512 dimensions; ALOE [2] uses 28 encoder layers with 1280 dimensions. Although one of the purposes of this paper is to highlight the advantage of OC representation in requiring less computational effort, at least a normal model scale should be considered: the current model scale setting, combined with the performance comparison between TF 2 and TF 5 models, raises doubts as to whether DINOv2 would perform better under a normal scale setting.

[1] MDETR-Modulated Detection for End-to-End Multi-Modal Understanding
[2] Attention over learned object embeddings enables complex visual reasoning

- **The uncertainty of generalization ability: ** Following up on the previous weakness, we can also easily identify another issue: this article aims to compare the "compositional generalization ability of visual feature", meaning that the generalization ability should be inherent in the visual features. However, in reality, the scale and capacity of the VQA model used largely influence the final measured generalization ability. For instance, when using TF 2, DINOSAURv2 demonstrates stronger generalization ability than DINOv2; but when using TF 5, DINOv2 shows stronger generalization ability than DINOSAURv2. So, which one is actually better at handling unseen objects? It seems difficult to make a thorough judgment through this experiment.

**Questions:**

- **Composition generalization and disentanglement: ** The compositional generalization in this paper refers to the model's ability to handle new objects formed by unknown compositions of known attributes. This seems to be related to the model's ability to decouple different attributes in objects. There are already similar studies in OCL, such as [3, 4]. If these methods that focus on decoupled representations are used to construct OC representations and applied to the VQA task, will there be better compositional generalization effects?

[3] Neural Systematic Binder
[4] Disentanglement via Latent Quantization

---

> ### Author Response · Authors · 2025-11-24
>
> We thank the reviewer for the careful reading and constructive in-depth feedback. We appreciate the positive assessment and recognition of the “methodological rigor”, “comprehensive experiments”, and the benchmark contribution as a “valuable resource for future research.” Additionally, we also highly appreciate the summary that our study “provides the first systematic, quantitative evidence that object-centric representations lead to stronger compositional generalization, especially under limited data, diversity, or compute conditions”. We address each of the raised concerns below. We refer to the general response for a summary of the central research question and a clarification of the design choices behind the controlled study of object‑centric (OC) representations, as well as references.
> ### “OC performs better in complex compositional generalization scenarios; dense features only outperform OC in simple scenarios.”
> We thank the reviewer for the detailed observations. We agree that CLEVRTex “medium” is a slight outlier in the DINOv2–based comparisons (Table 1). This is not the case for the SigLIP2-family (Table 11). We did not intend to claim a monotonically increasing OC advantage for generalization as difficulty grows (since this is not necessarily the case). The intended statement is that **dense features cease to outperform once compositional generalization is sufficiently difficult**. Concretely, DINOSAURv2 matches or slightly outperforms DINOv2 in all “hard” settings with both downstream models (6/6 comparisons). This is often already true for “medium” (4/6), with the main exceptions being Super‑CLEVR and MOVi‑C “medium” under TF 5; and it is rare for “easy” (2/6), where dense features are typically superior, when ignoring compute. We will refine this wording in the revision.
> ### Scale of VQA model
> We agree that a broader downstream capacity ablation is informative. Therefore, we conducted additional experiments that are shown below. We vary the number of layers (TF 10 and 15), the size of the hidden dimension (TF 5 h=256 and h=512), and both the model and hidden dimension (TF 5 d=265, h=1024). We would like to emphasize that none of the alternatives consistently outperform the “default” TF 5 downstream model (Table 1 and shown here), and most of the time they are worse, even though more compute is used. Additionally, the downstream model is sufficiently expressive to achieve 100% accuracy in‑distribution when given the “correct” image representation, i.e., ground‑truth object properties (Fig. 1; Sec. 4.1, Oracle).
> #### Downstream model ablations
> | Model            | Downstream          |   CLEVRTex EASY |   CLEVRTex MEDIUM |   CLEVRTex HARD |
> |------------------|---------------------|-----------------|-------------------|-----------------|
> | DINOv2           | TF 5                |            85.4 |              70.3 |            55.4 |
> | DINOSAURv2       | TF 5                |            82.5 |              73.3 |            55.5 |
> |------------------|---------------------|-----------------|-------------------|-----------------|
> | DINOv2           | TF 10               |            84.9 |              70.0 |            55.2 |
> | DINOSAURv2       | TF 10               |            81.3 |              70.7 |            53.5 |
> |------------------|---------------------|-----------------|-------------------|-----------------|
> | DINOv2           | TF 15               |            83.8 |              68.1 |            53.7 |
> | DINOSAURv2       | TF 15               |            81.6 |              71.9 |            54.3 |
> |------------------|---------------------|-----------------|-------------------|-----------------|
> | DINOv2           | TF 5 (h=256)        |            81.0 |              71.4 |            52.2 |
> | DINOSAURv2       | TF 5 (h=256)        |            81.1 |              72.6 |            53.8 |
> |------------------|---------------------|-----------------|-------------------|-----------------|
> | DINOv2           | TF 5 (h=512)        |            81.5 |              71.0 |            53.8 |
> | DINOSAURv2       | TF 5 (h=512)        |            81.7 |              69.8 |            53.7 |
> |------------------|---------------------|-----------------|-------------------|-----------------|
> | DINOv2           | TF 5 (d=256,h=1024) |            85.3 |              70.1 |            56.2 |
> | DINOSAURv2       | TF 5 (d=256,h=1024) |            81.1 |              68.2 |            54.5 |

---

> > ### Author Response · Authors · 2025-11-24
> >
> > ### Uncertainty of Generalization Ability
> > We thank the reviewer for their insightful comments. Evaluating visual features via a VQA task necessarily involves a downstream model. In our study, we **fix the downstream model family and ablate its size**, while varying only the type and size of the image representations. The picture is indeed nuanced: with the smaller model (TF 2), OC representations generalize consistently better (Tables 1, 11). To compare fairly across representation sizes and downstream model capacities, we analyze **generalization under the lens of downstream compute** (Sec. 4.3). TF 2 generalizes better than TF 5 within the same representation at low‑compute regimes, < 0.5 PFLOPS for DINOSAURv2 and < 1 PFLOPS for DINOv2 (Fig. 3). Across representations, DINOv2 requires more compute with TF 5 to surpass either downstream model paired with the OC features, especially on “hard” (Fig. 3). These results support our central claim: **OC features yield stronger compositional generalization when data, diversity, or compute are constrained**, while dense features can catch up or improve upon them given sufficient resources.
> > ## Questions
> > ### Compositional generalization and disentanglement
> > We agree that this is a valuable and interesting direction. Approaches that explicitly decouple attributes (e.g., the cited lines of work) could plausibly strengthen OC representations for compositional generalization. Integrating such methods while retaining our carefully controlled comparisons would, however, substantially expand the experimental grid. Additionally, we were most interested in the (compositional) generalization claims that were made in [2, 6, 7, 8] to study whether popular and competitive OC approaches (as reviewer zM3y notes: "foundational model + mature Slot Attention bottleneck") can generalize compositionally without further inductive biases (e.g., disentanglement).

---

### Official Review · Reviewer_EJ8F · 2025-10-30

**Soundness:** 2
**Presentation:** 2
**Contribution:** 2
**Rating:** 2
**Confidence:** 2

**Summary:**

This paper introduces a visual question answering (VQA) benchmark to validate whether object-centric (OC) visual representations yield better compositional generalization than dense (distributed) representations from visual encoders. The benchmark consists of three datasets (CLEVRTex, Super-CLEVR, and MOVi-C). For each base dataset, they further generate 3 training datasets by changing the diversity from “easy” to “hard”.  Dense encoders (DINOv2, SigLIPs) are compared with OC models (DINOSAURv2, SigLIPSAUR2), which use a Slot-Attention bottleneck. The goal is to evaluate the quality of representations using VQA on training sets of increasing difficulty.

Main findings:
- OC approaches are superior in harder compositional generalization settings
- OC shows better results across different budgets
- OC is more sample-efficient, achieving stronger generalization with less data.

However, the work’s conclusions rely on synthetic VQA benchmark under a distillation-style OC objective, yielding modest, regime-dependent gains that don’t convincingly establish OC representations as superior in realistic settings.

**Strengths:**

- Comprehensive experiments: Both in-distribution (ID) as well as compositional out-of-distribution (COOD) are reported. In COOD settings, they use 20% of object-property combinations for testing, while the rest for training.
- The paper is well written and easy to read.

**Weaknesses:**

- The results were only reported for synthetic datasets. No tests on natural images, real-image VQA, or open-vocabulary setups—so it’s unclear the findings transfer beyond those toy examples.
- Fig. 2 shows a strong ID-COOD correlation across settings. However, the ID–COOD correlation is somewhat expected, not novel (see [1] for example).  This is an unsurprising result that many would anticipate when training distributions are simplified. It doesn’t advance understanding of why dense and OC features differ.
- OC models are learned based on dense features. A pixel-space or alternative OC objective is not experimented.



[1] Miller, John P., et al. "Accuracy on the line: on the strong correlation between out-of-distribution and in-distribution generalization." International conference on machine learning. PMLR, 2021.

**Questions:**

- Figure 2 looks confusing. Why are there more than 4 points on each setting for each data set? There are only two configurations on image representation + 2 configurations on downstream models. Can you describe the figure in more detail?
- In training diversity, “object-centric (OC) representations degrade less and remain superior to dense features on harder generalizations”. However, the improvements are marginal when using the large downstream model TF 5 (see Table 11). Why?
- The feature dimensions of OC models  are different from their counterparts (DINOv2 of 384 vs DINOSAURv2 of 256 and SigLIP2 of 768 vs SigLIPSAUR2 of 256).  Why not set them to the same dimensionality?
- Missing description or description appeared after abbreviation, e.g., OC, CA, ODD.
- Fig. 1 is not referenced anywhere.

---

> ### Author Response · Authors · 2025-11-24
>
> We thank the reviewer for the detailed and constructive review. We’re grateful to read that our study offers “**comprehensive experiments**” and that “**The paper is well written and easy to read.**” We address the raised concerns below. We refer to the general response for a summary of the central research question and a clarification of the design choices behind the controlled study of object‑centric (OC) representations, as well as references.
> ## Weaknesses
> ### Scope and use of synthetic data.
> For a more detailed discussion, please refer to the general response (Framing and goal of the study & Differentiation of the proposed benchmark). The choice of synthetic data aligns with the paper’s goal: to cleanly test compositional generalization under full control of representation capacity, training data diversity, downstream compute, and sample size. Constructing splits that precisely hold out object-attribute combinations is challenging and often impossible to guarantee in natural-image corpora. Synthetic images, however, allow us to manipulate these axes without confounds. We agree that extending to natural-image VQA and open-vocabulary settings is valuable future work, but it falls outside the scope required for a controlled comparison in this study.
> ### ID-COOD correlation.
> We agree that a strong ID-COOD correlation can be expected and do not consider it a main contribution of our paper. We included it as a sanity check that prior observations also hold in our setting (similar to [3]). In general, the strong positive correlation can break down in some scenarios, as also noted by [4] and further studied in, for example, [5].
> ### Objective for OC models.
> The motivation for using a distillation-style OC objective from dense features is to control all axes for a fair and direct comparison to the same base encoder. We agree that exploring alternative OC objectives (including pixel-space) is interesting, but such approaches are typically worse in practice [2, 7] or would introduce additional confounds, making a fair comparison challenging. Additionally, we refer to the general response (Framing and goal of the study & Baseline choices) for a more detailed discussion.
> ## Questions
> ### Detailed description of Figure 2
> Figure 2 acts as a **visual summary of all models** and settings (Tables 7-10). For ease of comparison, refer to Table 1. For each training dataset (e.g., CLEVRTex Easy), there are six different image representations derived from either the original dense vision encoder (DINOv2) or its object-centric counterpart (DINOSAURv2). Then, each image representation is evaluated on the VQA task with either the two-layer (TF 2) or five-layer (TF 5) transformer downstream model. The same comparisons are made for the SigLIP2 vision encoder. Overall, this results in 2 (base vision encoders) x 6 (different image representations) x 2 (downstream models) = 24 data points for each training dataset (e.g., CLEVRTex Easy).
> ### Why are OC gains marginal with the larger downstream model?
> Dense representations benefit more from larger downstream models than their OC counterparts (Table 1). The relevant visual structure is harder to extract from dense features, so additional downstream model capacity helps to decrease the difference. We view **marginal improvements (or parity) as notable, given that dense encoders use more than three times the downstream compute** (Fig. 3).
> ### Feature dimensionalities
> We agree this would be an interesting direction. We chose commonly used embedding sizes for OC representations, as in [2, 7, 8]. Importantly, our comparison includes a **Cross-Attention (CA)** adapter that projects all image representations to the same size before the downstream model (for example, DINOv2 + CA vs. DINOSAURv2 in Table 1), ensuring an apples-to-apples comparison (Table 1, 6). We observe that the reduced, same-sized DINOv2 representation (DINOv2 + CA) performs worse than the object-centric representation of the same size (DINOSAURv2), as shown and described in Table 1 and Section 4.2 - Reduction with Cross-Attention.
> ### Late definitions of abbreviations & Figure 1 is not referenced
> We thank the reviewer for bringing this oversight to our attention. We will correct this in the revision.

---

### Official Review · Reviewer_iRSo · 2025-10-31

**Soundness:** 3
**Presentation:** 3
**Contribution:** 2
**Rating:** 2
**Confidence:** 4

**Summary:**

This study focuses on whether object-centric (OC) representations are more conducive to compositional generalization. By constructing a VQA benchmark and comparing mainstream visual encoders with their OC variants.

**Strengths:**

The paper conducts extensive experiments analyzing common visual encoders, offering us a deeper understanding of them.

**Weaknesses:**

1. How does the compositional generalization benchmark proposed in this paper fundamentally differ from existing compositional generalization tests? To my knowledge, other benchmarks also test compositional generalization with respect to attributes, e.g., cczsl [1] and c-gqa [2].

2. The authors propose three findings—what can we do with these findings? In other words, what insights do they provide? Why are these three findings important?

3. What is the underlying mechanism by which OC representations improve compositional generalization? Does the “object decomposition” in OC representations reduce the model’s reliance on “attribute co-occurrence frequencies” (whereas dense representations may overly depend on attribute co-occurrence patterns in the training set, leading to poor compositional generalization)?

4. It is commonly believed that slot attention is sensitive to the number and distribution of objects, yet the paper neither proposes improvements to address this limitation nor analyzes the impact of “variations in object number” on compositional generalization.

[1] Zhang Y, Feng S, Yuan J. Continual compositional zero-shot learning[C]//Proceedings of the Thirty-Third International Joint Conference on Artificial Intelligence. 2024: 1724-1732.

[2] Wang Q, Liu L, Jing C, et al. Learning conditional attributes for compositional zero-shot learning[C]//Proceedings of the IEEE/CVF conference on computer vision and pattern recognition. 2023: 11197-11206.

**Questions:**

see the weaknesses

---

> ### Author Response · Authors · 2025-11-24
>
> We thank the reviewer for the thoughtful and constructive review, and for the positive assessment that “The paper conducts extensive experiments analyzing common visual encoders, offering us a deeper understanding of them.” We address each of the concerns raised below. We refer to the general response for a summary of the central research question and clarification of the design choices behind the controlled study of object‑centric (OC) representations, as well as references.
> ### 1. How our benchmark differs from prior compositionality benchmarks.
> For a more general discussion, we refer to the general response (Differentiation of the proposed benchmark). C-GQA is similar in spirit, but makes it difficult to study multi-object, multi-attribute relationships in a balanced and controlled setting where the difficulty of generalization can be controlled by **adjusting training diversity without reducing the number of samples**. In contrast to arguably the most similar benchmark for testing compositional generalisation, i.e., [1], the benchmark in this study (Fig. 1; App. A) uses (1) **more visually complex images**; CLEVRTex is the simplest setting here versus the most complex there. (2) We introduce **novel VQA task formulations** for CLEVRTex and MOVi‑C that test compositional generalisation with **up to six objects (vs. two)** and **far more attribute combinations** (**up to 2,688** in Super‑CLEVR, even excluding part–whole relationships vs. **192**).
> ### 2. What can we do with the three findings?
> We refer to the general response for the motivation of studying compositional generalization of object-centric representations (Framing and goal of the study). The study examines compositional generalization along three practically relevant axes: **training data diversity, compute, and sample size**. The actionable takeaway is that object-centric (OC) representations offer stronger compositional generalization when any one of these resources is constrained, whereas dense representations match or surpass OC models only when provided with sufficient data, diversity, and downstream compute. This provides concrete scenarios for practitioners and, to our knowledge, is the **first empirical demonstration of an OC advantage in compositional generalisation under a precise and controlled setting** (as highlighted by reviewer oVPf).
> ### 3. Mechanism of better compositional generalisation
> We agree that a complete mechanistic answer is a very interesting research question, but argue that it is beyond the present scope. We speculate, however, that OC decompositions help because they separate the scene into object-wise sub-representations that share a common format. This makes the relevant information easier to extract and likely reduces reliance on spurious attribute co-occurrences compared to dense representations.

---

> > ### Author Response · Authors · 2025-11-24
> >
> > ### 4. Sensitivity of slot attention to the number of objects.
> > We agree that slot-based models can be sensitive to the number of objects. The goal of this study is to compare representational biases under controlled conditions; improvements to the OC architecture are orthogonal to this goal. We have conducted **additional experiments** by varying the **number of objects at test time** (shown below) and the **number of slots at training time** (discussed in the answer “1. Choice of number of slots” for reviewer zM3y), to investigate the effect on compositional generalisation and to demonstrate that the main conclusions still hold under reasonable ranges. When testing the compositional generalization on a novel CLEVRTex COOD test set with 7-10 objects, i.e., more than at training time, OC representations still generalize better compositionally with TF 2 compared to dense counterparts but are slightly surpassed by the dense representations with TF 5, although the differences for harder generalizations are usually small. We would like to note that this test scenario is complex: both the representation (7 slots) and the downstream model (7 tokens) for the object-centric models have not dealt with more tokens (11 at test time), while the dense counterparts always use the same number of large tokens at train and test time.
> > #### CLEVRTex COOD: more objects
> > | Model + TF 2   |   CLEVRTex EASY |   CLEVRTex MEDIUM |   CLEVRTex HARD |
> > |----------------|-----------------|-------------------|-----------------|
> > | DINOv2         |            61.1 |              50.4 |            44.6 |
> > | DINOSAURv2     |            65.8 |              60.5 |            47.9 |
> > | SigLIP2        |            64.4 |              56.6 |            45.4 |
> > | SigLIPSAUR2    |            72.0 |              64.4 |            49.6 |
> >
> > | Model + TF 5   |   CLEVRTex EASY |   CLEVRTex MEDIUM |   CLEVRTex HARD |
> > |----------------|-----------------|-------------------|-----------------|
> > | DINOv2         |            75.7 |              62.3 |            49.2 |
> > | DINOSAURv2     |            72.2 |              62.4 |            47.9 |
> > | SigLIP2        |            78.4 |              66.1 |            50.9 |
> > | SigLIPSAUR2    |            74.6 |              64.1 |            50.7 |

---

### Official Review · Reviewer_zM3y · 2025-11-01

**Soundness:** 3
**Presentation:** 2
**Contribution:** 2
**Rating:** 4
**Confidence:** 4

**Summary:**

This paper investigates the role of object-centric (OC) representations in achieving compositional generalization in visual question answering (VQA) tasks.  The authors introduce a benchmark that evaluates how well vision models, with and without object-centric biases, generalize to unseen combinations of object properties.

**Strengths:**

1. The paper is well-organized, with a logical flow from the introduction to the experimental design, making it easy for readers to follow the authors' reasoning and methodology.
2. The experimental part is relatively thorough and comprehensive.

**Weaknesses:**

（1）Limited Novelty in Methodology: The core object-centric (OC) models proposed in this paper (DINOSAURv2, SigLIPSAUR2) are essentially derivative models built upon existing foundational models (DINOv2, SigLIP2) combined with Slot Attention (Locatello et al., 2020).   No novel object decomposition mechanism or representation learning paradigm is introduced.   The technical approach is essentially a combination of "existing foundational models + mature Slot Attention bottleneck," which overlaps heavily with the design of DINOSAURv2 by Didolkar et al. (2024).   There is no breakthrough in the core principles of object-centric representations

（2）Lack of Innovation in Benchmark Design: The VQA benchmark proposed in the paper, which includes CLEVRTex, Super-CLEVR, and MOVi-C, follows the same data generation logic as Kim et al. (2024) with its “attribute combination partition + training/test splits” approach.  The difficulty of the generalization task is controlled simply by adjusting the proportion of object-attribute combinations in the training set (e.g., "easy", "medium", "hard" corresponding to 80%/40%/20% of the combinations).  No new dimensions of generalization challenges are introduced, such as object occlusion, or real-world noise.  Compared to existing synthetic datasets like ConceptMix and SugarCrepe, this benchmark lacks differentiation in design.

（3）Limited Model Comparison: The paper compares the OC models only against the original dense representations (DINOv2, SigLIP2) and k-means clustering representations but overlooks non-OC baseline[1] models that currently perform well on compositional generalization tasks.
 [1] Kempf E, Schrodi S, Argus M, et al. When and How Does CLIP Enable Domain and Compositional Generalization?[J]. arXiv preprint arXiv:2502.09507, 2025.

**Questions:**

(1) In Section 3.2, "Models and Evaluation", the Vision Models section mentions that the object-centric feature dimension is set to 7×256. Why was the value of 7 chosen for the number of slots? Would varying the number of slots affect the performance of the experiments?

(2) In Section 3.2, it is mentioned that the OC model (e.g., DINOSAURv2) requires additional training to “reconstruct dense features from the foundation model,” essentially performing a "secondary refinement" of the base model features. Could this process allow the model to learn structured object information prematurely, rather than this being an advantage purely of the "OC representation" itself

(3) In the experimental section, the paper only evaluates models on the custom-built datasets. Have the authors considered evaluating the object-centric representations on more complex, real-world datasets, such as GQA[1]? These types of life-like, more intricate datasets could provide further insights into the effectiveness of object-centric representations for compositional generalization tasks.

[1] Hudson D A, Manning C D. Gqa: A new dataset for real-world visual reasoning and compositional question answering[C]//Proceedings of the IEEE/CVF conference on computer vision and pattern recognition. 2019: 6700-6709.

---

> ### Author Response · Authors · 2025-11-24
>
> We thank the reviewer for the careful read and constructive feedback. We are particularly grateful for the positive remarks that “The paper is well-organized, with a logical flow from the introduction to the experimental design, making it easy for readers to follow the authors' reasoning and methodology”, and that “The experimental part is relatively thorough and comprehensive.” Below, we address the comments in detail. Additionally, please refer to the general response for a summary of the central research question and clarification of the design choices behind the controlled study of object-centric (OC) representations, as well as references.
>
> ## Weaknesses
> ### 1. Limited Novelty in Methodology
> We agree that the OC models (DINOSAURv2, SigLIPSAUR2) are derivatives of strong foundation models paired with a mature OC bottleneck, not a completely novel design, which we acknowledge explicitly. The goal of the study is not to introduce a new object decomposition mechanism, but to **measure the isolated effect of the OC inductive bias under matched backbone, capacity, and compute.** For a more detailed discussion, please refer to the general response (Framing and goal of the study & Baseline choices).
> ### 2. Lack of Innovation in Benchmark Design
> We agree with the reviewer that the benchmark draws inspiration from Kim et al. (2024) [1]. Refer to the general response for a summary of the primary goal of this study (Framing and goal of the study) and additional differentiations of the benchmark (Differentiation of the proposed benchmark). The benchmark design differs in the following ways from Kim et al. (2024) [1], which are important for isolating compositional effects for visually rich scenes and complex tasks. First, our visual worlds contain more complex images. CLEVRTex, for example, is the most visually complex dataset in Kim et al. (2024) [1] but serves as our simplest. Second, we introduce novel VQA task formulations for CLEVRTex and MOVi-C that test compositional generalisation in multi-object (up to six objects compared to only two) and multi-attribute (e.g., up to 2688 combinations for Super-CLEVR, excluding part-whole combinations, versus 192) reasoning within a controlled pipeline. Third, although we do not isolate occlusion as an explicit axis, occlusions are naturally present in the images (Fig. 1; Appendix A).
>
> Regarding “real‑world noise”, we would appreciate clarification on which factors the reviewer has in mind. As one possible example of real-world noise we consider unseen (OOD) backgrounds, where we present the results of new experiments on a MOVi-C COOD (images) test set with unseen backgrounds below. We make two observations: 1) All previous observations still hold; among others, OC representations are better for harder compositional tasks. 2) Overall, the accuracies are very close to the MOVi-C COOD test set without OOD background (compare Tables 8, 10), indicating that the main difficulty in the task is due to novel objects.
>
> #### MOVi-C COOD + OOD Background
> | Model + TF 2   |   MOVi-C EASY |   MOVi-C MEDIUM |   MOVi-C HARD |
> |----------------|---------------|-----------------|---------------|
> | DINOv2         |          57.7 |            53.7 |          51.6 |
> | DINOSAURv2     |          58.3 |            55.0 |          53.2 |
> | SigLIP2        |          58.3 |            54.6 |          53.2 |
> | SigLIPSAUR2    |          61.4 |            57.1 |          54.9 |
>
> | Model + TF 5   |   MOVi-C EASY |   MOVi-C MEDIUM |   MOVi-C HARD |
> |----------------|---------------|-----------------|---------------|
> | DINOv2         |          60.1 |            56.3 |          54.3 |
> | DINOSAURv2     |          59.3 |            56.0 |          54.4 |
> | SigLIP2        |          61.6 |            57.8 |          55.1 |
> | SigLIPSAUR2    |          61.4 |            57.9 |          55.5 |
>
> The aim of the study is to maintain a controlled setup such that any gains can be attributed to representational bias rather than confounds. Finally, we want to highlight that SugarCrepe and ConceptMix, although valuable and acknowledged in the Related Work section, are not suitable for directly evaluating vision representations under tight control (neither accounts for train vs. test differences, i.e., they only provide a test set), consist of a different task (ConceptMix: image generation; SugarCrepe: image-to-text retrieval), and/or target a different notion of compositionality (SugarCrepe tests compositional understanding of the correct text to the image).

---

> > ### Author Response · Authors · 2025-11-24
> >
> > ### 3. Limited Model Comparison
> > We agree with this observation. To answer the main research question of the study (refer to the general response for more details), we intentionally restrict comparisons to dense counterparts of the same foundations and to size-matched non-OC reductions, thereby minimizing confounding factors. Comparing to non-OC baselines that differ in data, objectives, or model size would make it hard to tell whether any differences in generalization are due to the inductive bias of using an object-centric representation.
> > ## Questions
> > ### 1. Choice of number of slots
> > For controlled benchmarks, a common choice of OC representations is to set the number of slots to the maximum number of objects per image plus one background slot. This is the reason for choosing 7 slots [2, 6]. We include new experiments below that ablate the number of slots and notice that 7 slots are often not the best choice for generalization; instead, using a few more slots (9 or 11) is usually better. This further strengthens the findings (compare Table 1), as the additional compute from using a few more slots is negligible compared to the larger representation of the dense counterpart (DINOv2: 256 tokens), and this further closes or increases the gap.
> > #### CLEVRTex COOD: Number of Slots
> > | Model + TF 2          |  Number of slots  |  CLEVRTex EASY  |  CLEVRTex MEDIUM  |  CLEVRTex HARD  |
> > |-----------------------|-------------------|-----------------|-------------------|-----------------|
> > | DINOSAURv2            |         5         |      68.2       |       63.6        |      51.2       |
> > | DINOSAURv2            |         7         |      76.5       |       71.2        |      55.6       |
> > | DINOSAURv2            |         9         |      80.0       |       72.4        |      55.1       |
> > | DINOSAURv2            |        11         |      79.7       |       73.6        |      56.3       |
> >
> > | Model + TF 5          |  Number of slots  |  CLEVRTex EASY  |  CLEVRTex MEDIUM  |  CLEVRTex HARD  |
> > |-----------------------|-------------------|-----------------|-------------------|-----------------|
> > | DINOSAURv2            |         5         |      71.9       |       65.3        |      51.0       |
> > | DINOSAURv2            |         7         |      82.5       |       73.3        |      55.5       |
> > | DINOSAURv2            |         9         |      83.6       |       74.1        |      56.0       |
> > | DINOSAURv2            |        11         |      85.7       |       74.3        |      56.8       |
> >
> > ### 2. Do the benefits come from the OC module or the secondary refinement?
> > We agree that the OC module receives an additional training stage where it is optimized to reconstruct dense features from a foundation model. However, our experiments show that the performance gains are not solely due to this secondary refinement. In particular, in all experiments, we use two alternatives to Slot Attention: Cross-Attention (CA) trained in a similar manner, and k-means applied on the backbone features. Under these matched conditions, we find that OC representations yield stronger compositional generalisation than CA or k-means at the same size (Table 1).
> > ### 3. Why not evaluate on real‑world datasets such as GQA?
> > We agree that such evaluations are valuable. However, we would like to highlight two points:
> > - Real-world datasets, such as GQA, do not allow for the same fully controlled setting required for the precise definition of compositional generalization studied here. The focus is on a clear attribution to representational bias under matched capacity, data, and compute. We therefore consider evaluations like GQA to be orthogonal to our study.
> > - The effectiveness of OC representations on real-world datasets such as GQA has already been studied [2]. OC models are evaluated on real-world datasets, such as GQA, and found to perform on par or better than their foundation model counterparts. The goal of this paper is therefore not to reestablish those results, but to complement them: we focus on fully controlled settings where object attributes and compositions can be systematically manipulated, in order to isolate and analyze the specific advantages and failure modes of OC representations.

---

### Author Response · Authors · 2025-11-24
**General Response**

We thank all reviewers for their helpful feedback. We are happy to see that the reviewers find our paper well-written (zM3y, EJ8F), with comprehensive experimentation (zM3y, iRSo, EJ8F, oVPF), in a carefully controlled and methodologically rigorous setup (oVPF).

Common concerns were raised regarding 1) the precise goal of the study, 2) the baseline choices, and 3) the differentiation of our proposed compositional VQA benchmark relative to existing benchmarks. We address these points below.

- **Framing and goal of the study.** The paper’s primary aim is to investigate how well object-centric (OC) representations handle compositional generalization, as this is often cited as one of their main motivations [2, 6]. To enable a scientifically sound comparison, we employ a fully controlled setup in which we systematically vary the following axes: representation capacity, training data diversity, downstream compute, and sample size. Then, we directly compare OC variants with their dense counterparts. As reviewer oVPf nicely summarizes: “the study provides the first systematic, quantitative evidence that object-centric representations lead to stronger compositional generalization, especially under limited data, diversity, or compute conditions”.
- **Baseline choices.** To ensure this fair comparison of object-centric representations, we require a direct comparison with the dense representations of strong, pretrained vision encoders and a competitive candidate for object-centric representations. Therefore, we chose as reviewer zM3y points out "existing foundational model + mature Slot Attention bottleneck" to fulfill both of these needs [2, 6, 7, 8]. Additionally, by varying over all additional axes (representation capacity, training data diversity, downstream model, sample size), the overall grid of experiments becomes quite large quickly (over 70k GPU-hours for this study), which further justifies the choice of popular and competitive OC approaches.
- **Differentiation of the proposed benchmark.** We define compositional generalization as the ability to reason about novel combinations of familiar concepts (e.g., shape, color, material, size) [1, 3]. To study this phenomenon cleanly, a benchmark must provide precise control over which attribute combinations appear in training vs. test. Existing real-world benchmarks, such as (C-)GQA, do not allow for full control over scene compositions. Therefore, we design a novel compositional VQA benchmark on synthetic datasets (CLEVRTex, Super-CLEVR, MOVi-C), where we have full control over object attributes and their compositions, and can systematically construct ID/COOD splits at different difficulty levels.

We will further address other concerns and questions in the responses to each review, which include new experiments for 1) the number of slots for OC representations, 2) compositional generalization to more objects, 3) additional downstream models, and 4) unseen backgrounds.

---

> ### Author Response · Authors · 2025-11-24
>
> References.
>
> [1] Kim, Y., Singh, G., Park, J., Gulcehre, C., & Ahn, S. (2023). Imagine the unseen world: a benchmark for systematic generalization in visual world models. Advances in Neural Information Processing Systems, 36, 27880-27896.
>
> [2] Mamaghan, A. M. K., Papa, S., Johansson, K. H., Bauer, S., & Dittadi, A. (2024). Exploring the Effectiveness of Object-Centric Representations in Visual Question Answering: Comparative Insights with Foundation Models. In The Thirteenth International Conference on Learning Representations.
>
> [3] Abbasi, R., Rohban, M. H., & Baghshah, M. S. (2024). Deciphering the Role of Representation Disentanglement: Investigating Compositional Generalization in CLIP Models. In European Conference on Computer Vision (pp. 35-50).
>
> [4] Miller, J. P., Taori, R., Raghunathan, A., Sagawa, S., Koh, P. W., Shankar, V., Liang, P., Carmon, Y., & Schmidt, L. (2021, July). Accuracy on the line: on the strong correlation between out-of-distribution and in-distribution generalization. In International conference on machine learning (pp. 7721-7735). PMLR.
>
> [5] Sanyal, A., Hu, Y., Yu, Y., Ma, Y., Wang, Y., & Schölkopf, B. (2025). Accuracy on the wrong line: On the pitfalls of noisy data for out-of-distribution generalisation. In International Conference on Artificial Intelligence and Statistics (pp. 2170-2178). PMLR.
>
> [6] Locatello, F., Weissenborn, D., Unterthiner, T., Mahendran, A., Heigold, G., Uszkoreit, J., Dosovitskiy, A., & Kipf, T. (2020). Object-centric learning with slot attention. Advances in neural information processing systems, 33, 11525-11538.
>
> [7] Seitzer, M., Horn, M., Zadaianchuk, A., Zietlow, D., Xiao, T., Simon-Gabriel, C. J., He, T., Zhang, Z., Schölkopf, B., Brox, T., & Locatello, F. Bridging the Gap to Real-World Object-Centric Learning. In The Eleventh International Conference on Learning Representations.
>
> [8] Didolkar, A. R., Zadaianchuk, A., Goyal, A., Mozer, M. C., Bengio, Y., Martius, G., & Seitzer, M. (2025). On the transfer of object-centric representation learning. In The Thirteenth International Conference on Learning Representations.

---

### Meta-Review · Area_Chair_jLBs · 2025-12-18

**Summary:**

This paper presents an empirical study on the compositional generalization of object-centric (OC) versus dense visual representations. To this end, the authors introduce a new synthetic benchmark and systematically compare the generalization performance of OC and dense representations using visual question answering (VQA) as a downstream task.

The paper initially received two weak-reject (4) and two reject (2) recommendations. The primary concerns raised by the reviewers can be summarized as follows:

- **Limited novelty (zM3y, iRSo, EJ8F):** Both the modeling choices (e.g., foundation models combined with slot attention) and the empirical analyses are considered incremental. While the benchmark includes more visually complex scenes and the VQA-based evaluation provides useful additional evidence, the datasets remain fully synthetic and the overall evaluation protocol substantially overlaps with prior work (e.g., Kim et al., 2024).
- **Limited evaluation scope (EJ8F, zM3y):** The empirical results are restricted to synthetic environments. The rebuttal reasonably motivates the use of synthetic data for controlled compositional analysis; however, it does not provide convincing insight into how these findings would translate to realistic settings, where notions of “objectness” and the boundary between in-distribution and out-of-distribution compositions are not as clearly defined as in synthetic benchmarks.
- **Mismatch between claims and empirical evidence (oVPf, EJ8F):** Especially with larger downstream models, the advantages of OC representations are small or mixed, making broad claims of superiority difficult to sustain. While the rebuttal proposes refined wording and additional ablations, these changes may not be sufficient to fully resolve the concern, as the core claims remain only partially supported by the reported results.
- **Potential unfairness in the comparison protocol:** The OC modules are trained using in-domain data from the evaluation benchmarks, which introduces domain-specific inductive biases (e.g., objectness tailored to each domain), whereas the dense foundation models are kept frozen. To more cleanly isolate the effect of object-centric representations from domain adaptation effects, the OC modules would ideally be trained on held-out pre-training data rather than on the evaluation domains themselves.

Overall, while the work is thorough and the controlled benchmark may be useful to the community, the rebuttal does not sufficiently address the core concerns to overturn the original reviewer assessments. Therefore, the AC recommends rejection this time.

**Reviewer Concerns:**

see above

**Reviewer Scores:**

see above

---

### Decision · Program_Chairs · 2026-01-26

Reject